# A synergistic workspace for human consciousness revealed by Integrated Information Decomposition

**Andrea I Luppi[1,2]\*, Pedro AM Mediano[3†], Fernando E Rosas[4,5,6‡], Judith Allanson[1,7], John Pickard[1,8,9], Robin L Carhart-Harris[4,10], Guy B Williams[1,8], Michael M Craig[1,2], Paola Finoia[1], Adrian M Owen[11], Lorina Naci[12], David K Menon[2,8], Daniel Bor[3§], Emmanuel A Stamatakis[2]**

[1]Department of Clinical Neurosciences, University of Cambridge, Cambridge, United Kingdom; [2]University Division of Anaesthesia, School of Clinical Medicine, University of Cambridge, Cambridge, United Kingdom; [3]Department of Psychology, University of Cambridge, Cambridge, United Kingdom; [4]Center for Psychedelic Research, Department of Brain Science, Imperial College London, London, United Kingdom; [5]Center for Complexity Science, Imperial College London, London, United Kingdom; [6]Data Science Institute, Imperial College London, London, United Kingdom; [7]Department of Neurosciences, Cambridge University Hospitals NHS Foundation, Addenbrooke's Hospital, Cambridge, United Kingdom; [8]Wolfson Brain Imaging Centre, University of Cambridge, Cambridge, United Kingdom; [9]Division of Neurosurgery, School of Clinical Medicine, University of Cambridge, Addenbrooke's Hospital, Cambridge, United Kingdom; [10]Psychedelics Division - Neuroscape, Department of Neurology, University of California, San Francisco, United States; [11]Department of Psychology and Department of Physiology and Pharmacology, The Brain and Mind Institute, University of Western Ontario, London, Canada; [12]Trinity College Institute of Neuroscience, School of Psychology, Lloyd Building, Trinity College, Dublin, Ireland

**\*For correspondence:**
al857@cam.ac.uk

**Present address:** †Department of Computing, Imperial College London, London, United Kingdom; ‡Department of Informatics, University of Sussex, Brighton, United Kingdom; §Queen Mary University of London, London, United Kingdom

**Abstract** How is the information-processing architecture of the human brain organised, and how does its organisation support consciousness? Here, we combine network science and a rigorous information-theoretic notion of synergy to delineate a 'synergistic global workspace', comprising gateway regions that gather synergistic information from specialised modules across the human brain. This information is then integrated within the workspace and widely distributed via broadcaster regions. Through functional MRI analysis, we show that gateway regions of the synergistic workspace correspond to the human brain's default mode network, whereas broadcasters coincide with the executive control network. We find that loss of consciousness due to general anaesthesia or disorders of consciousness corresponds to diminished ability of the synergistic workspace to integrate information, which is restored upon recovery. Thus, loss of consciousness coincides with a breakdown of information integration within the synergistic workspace of the human brain. This work contributes to conceptual and empirical reconciliation between two prominent scientific theories of consciousness, the Global Neuronal Workspace and Integrated Information Theory, while also

advancing our understanding of how the human brain supports consciousness through the synergistic integration of information.

## eLife assessment

This article presents **important** results describing how the gathering, integration, and broadcasting of information in the brain changes when consciousness is lost either through anesthesia or injury. They provide **convincing** evidence to support their conclusions, although the paper relies on a single analysis tool (partial information decomposition) and could benefit from a clearer explication of its conceptual basis, methodology, and results. The work will be of interest to both neuroscientists and clinicians interested in basic and clinical aspects of consciousness.

## Introduction

Humans and other vertebrates rely on a centralised nervous system to process information from the environment, obtained from a wide array of sensory sources. Information from different sensory sources must eventually be combined - and integrated - with the organism's memories and goals, in order to guide adaptive behaviour effectively (*Varela et al., 2001*). However, understanding how the brain's information-processing architecture enables the integration of information remains a key open challenge in neuroscience (*Petersen and Sporns, 2015*; *Shine, 2019*). Theoretical and empirical work in cognitive neuroscience indicates that information processed in parallel by domain-specific sensory modules needs to be integrated within a multimodal 'central executive' (*Fodor, 1985*). Indeed, recent work has identified subsets of regions that are consistently recruited across a variety of tasks (*Deco et al., 2021b*; *Assem et al., 2020*; *Shine et al., 2019*), situated at the convergence of multiple anatomical, functional, and neurochemical hierarchies in the brain (*Hagmann et al., 2008*; *Felleman and Van Essen, 1991*; *Goulas et al., 2021*; *Sydnor et al., 2021*; *Baum et al., 2020*; *Bertolero et al., 2017*; *Vázquez-Rodríguez et al., 2019*; *Margulies et al., 2016*; *Burt et al., 2018*; *Demirtaş et al., 2019*; *Deco et al., 2021a*; *Hansen et al., 2021*; *Hansen et al., 2022*).

Prominent theories in cognitive and computational neuroscience have also proposed that global integration of information from diverse sources plays a fundamental role in relation to human consciousness (*Tononi et al., 2016*; *Seth and Bayne, 2022*). The influential Global Neuronal Workspace Theory (GNWT) focuses on the process by which specific neural information becomes available for conscious access, as occurring through the global integration induced by a 'global workspace' (*Dehaene and Changeux, 2011a*; *Mashour et al., 2020*; *Dehaene et al., 2011b*; *Baars, 2005*). Within the workspace, relevant information from different sources is integrated and subsequently broadcasted back to the entire brain, in order to inform further processing and achieve 'experiential integration' of distributed cortical modules into a coherent whole (*Mashour et al., 2020*; *Dehaene et al., 2011b*; *Dehaene and Naccache, 2001*). Thus, the global workspace is attributed both the role of integrator, and the role of orchestrator of cognitive function. Also highlighting the importance of integration, the prominent Integrated Information Theory (IIT; *Tononi et al., 2016*; *Tononi, 2008*; *Tononi, 2004*) posits that the degree of consciousness in a system is determined by its 'integrated information': the amount of intrinsic information generated by the dynamics of the system considered as a whole, over and above the information generated by the dynamics of its individual constituent parts (*Tononi et al., 2016*; *Tononi, 2008*; *Tononi, 2004*; *Tononi et al., 1998*). Thus, this notion of integrated information corresponds to the extent to which 'the whole is greater than the sum of its parts' (*Balduzzi and Tononi, 2008*).

Therefore, leading theoretical accounts of consciousness converge on this point: consciousness critically depends on the capability for global integration across a network of differentiated modules. Despite agreeing on the fundamental importance of information integration (*Cavanna et al., 2018*), these theories differ on its specific role and corresponding neural mechanisms. In contrast to GNWT's account, whereby integration is viewed as a necessary – but not sufficient – prerequisite step on the way to broadcasting and consciousness, IIT proposes a more fundamental identity between consciousness and the integration of information, but without specifying a formal architecture for this process: that is, according to IIT any system that integrates information will thereby be conscious, regardless of its specific organisation (*Balduzzi and Tononi, 2008*). Seen under this light, it becomes apparent that

**eLife digest** The human brain consists of billions of neurons which process sensory inputs, such as sight and sound, and combines them with information already stored in the brain. This integration of information guides our decisions, thoughts, and movements, and is hypothesized to be integral to consciousness. However, it is poorly understood how the brain regions responsible for processing this integration are organized in the brain.

To investigate this question, Luppi et al. employed a mathematical framework called Partial Information Decomposition (PID) which can distinguish different types of information: redundancy (available from many regions) and synergy (which reflects genuine integration). The team applied the PID framework to the brain scans of 100 individuals. This allowed them to identify which brain regions combine information from across the brain (known as gateways), and which ones transmit it back to the rest of the brain (known as broadcasters).

Next, Luppi et al. set out to find how these regions compared in unconscious and conscious individuals. To do this, they studied 15 healthy volunteers whose brains were scanned (using a technique called functional MRI) before, during, and after anaesthesia. This revealed that the brain integrated less information when unconscious, and that this reduction happens predominantly in gateway rather than broadcaster regions. The same effect was also observed in the brains of individuals who were permanently unconscious due to brain injuries.

These findings provide a way of understanding how information is organised in the brain. They also suggest that loss of consciousness due to brain injuries and anaesthesia involve similar brain circuits. This means it may be possible to gain insights about disorders of consciousness from studying how people emerge from anaesthesia.

IIT and GNWT are actually addressing different aspects of consciousness, and their views of integration are different but potentially complementary.

Crucially, our ability to make sense of any information-processing architecture is limited by our understanding of the information that is being processed. An elegant formal account of information in distributed systems – such as the human brain – is provided by the framework of Partial Information Decomposition (PID; *Williams and Beer, 2010*) which extends the formalism of Shannon mutual information by demonstrating that not all information is equal. Mutual information quantifies the reduction in uncertainty about one variable, when another variable is taken into account. In the case when more than one source of information is present, PID demonstrates that two sources can possess information about a given target that is unique (each source provides independent information), redundant (the same information is provided by both sources) or synergistic (complementary information, a higher order kind of information that is available only when both sources are considered together). As an example, humans have two sources of visual information about the world: two eyes. The information that is lost when one eye is closed is called the 'unique information' of that source – information that cannot be obtained from the remaining eye. The information that one still has when one eye is closed is called 'redundant information' – because it is information that is carried equally by both sources. This provides robustness: you can still see even after losing one eye. However, losing one eye also deprives you of stereoscopic information about depth. This information does not come from either eye alone: you need both, in order to perceive the third dimension. Therefore, this is called the 'synergistic information' between the sources – the extra advantage that is derived from combining them. Synergistic information therefore reflects the meaning of integration-as-cooperation, whereby elements are distinct from each other, but complementary (*Luppi et al., 2024a*).

Adding to the rich literature that addresses neural information from the perspective of encoding and decoding of task variables (*Quian Quiroga and Panzeri, 2009*,) there is growing appreciation that distinct types of information – as identified by information decomposition – may play a key role in the distributed information-processing architecture of the brain (*Luppi et al., 2024a*; *Timme et al., 2014*; *Sherrill et al., 2021*; *Faber et al., 2019*; *Sherrill et al., 2020*; *Newman et al., 2022*; *Celotto et al., 2023*; *Varley, 2023*; *Varley et al., 2023a*). Information decomposition can be applied to neural data from different scales, from electrophysiology to functional MRI, with or without reference to behaviour (*Luppi et al., 2024a*). When behavioural data are taken into account, information

decomposition can shed light on the processing of 'extrinsic' information, understood as the translation of sensory signals into behavioural choices across neurons or regions (*Celotto et al., 2023*; *Varley et al., 2023a*; *Delis et al., 2022*; *Francis et al., 2022*). However, information decomposition can also be applied to investigate the 'intrinsic' information that is present in the brain's spontaneous dynamics in the absence of any tasks, in the same vein as resting-state 'functional connectivity' and methods from statistical causal inference such as Granger causality (*Barnett et al., 2009*). In this context, information processing should be understood in terms of the dynamics of information: where and how information is stored, transferred, and modified (*Luppi et al., 2024a*). Specifically, since the future state of the brain is at least in part determined by its previous state, it is possible to view the future state of neural units (be they regions or neurons) as the target, and ask how it is determined by the same units' previous state, and the previous state of other units, which become the sources of information. Then, redundancy between two units occurs when their future spontaneous evolution is predicted equally well by the past of either unit. Synergy instead occurs when considering the two units together increases the mutual information between the units' past and their future – suggesting that the future of each is shaped by its interactions with the other. At the microscale (e.g. for spiking neurons) this phenomenon has been suggested as reflecting 'information modification' (*Timme et al., 2014*; *Newman et al., 2022*; *Wibral et al., 2014*). Synergy can also be viewed as reflecting the joint contribution of parts of the system to the whole, that is not driven by common input (*Mediano et al., 2018*).

By applying a recent generalisation of PID for timeseries data – known as Integrated Information Decomposition (*Varley, 2023*; ; *Mediano et al., 2021*) – we developed an information-resolved approach to decompose the information carried by brain dynamics and their intrinsic fluctuations (*Luppi et al., 2022b*). Traditional measures of statistical association ('functional connectivity') cannot disentangle synergy and redundancy; in fact, recent work has demonstrated that functional connectivity predominantly reflects redundant interactions (*Luppi et al., 2024a*; *Luppi et al., 2022b*; *Varley et al., 2023b*). In contrast, applying our information-resolved framework to functional MRI recordings of the human brain revealed that different regions of the human brain predominantly rely on different kinds of information for their interactions with other regions. Through this approach, we identified a 'synergistic core' of brain regions supporting higher level cognitive functions in the human brain through the synergistic integration of information (*Luppi et al., 2022b*). Similar results of a synergistic architecture were recently and independently obtained using a different decomposition (based on entropy rather than mutual information) (*Varley et al., 2023b*).

We also observed that a synergy-based measure of emergent dynamics in functional MRI recordings is disrupted in patients suffering from chronic disorders of consciousness (*Luppi et al., 2023a*). Building on these findings, it is natural to ask whether this synergistic core could correspond to the brain's global workspace. Furthermore, given that the views on information integration put forward by GNWT and IIT are potentially complementary, an important challenge to move the field forward is to leverage both accounts into a unified architecture that could explain empirical effects observed in neuroimaging data.

Therefore, this work sets out to address two fundamental questions of contemporary neuroscience:

1. How is the cognitive architecture of the human brain functionally organised, from an information-theoretic standpoint? Specifically, what brain regions does it involve, and what are the roles of the two kinds of information integration proposed by GNWT and IIT within this architecture?
2. How are different types of information in the brain related to human consciousness?

To address these questions, and provide an information-resolved view of human consciousness, here we study three resting-state fMRI datasets: (i) N=100 subjects from the Human Connectome Project; (ii) N=15 healthy volunteers who were scanned before and after general anaesthesia with the intravenous propofol as well as during post-anaesthetic recovery (*Luppi et al., 2019*) (iii) N=22 patients suffering from chronic disorders of consciousness (DOC) as a result of severe brain injury (*Luppi et al., 2019*). By comparing functional brain scans from transient anaesthetic-induced unconsciousness and from the persistent unconsciousness of DOC patients, which arises from brain injury, we can search for common brain changes associated with loss of consciousness – thereby disambiguating what is specific to loss of consciousness.

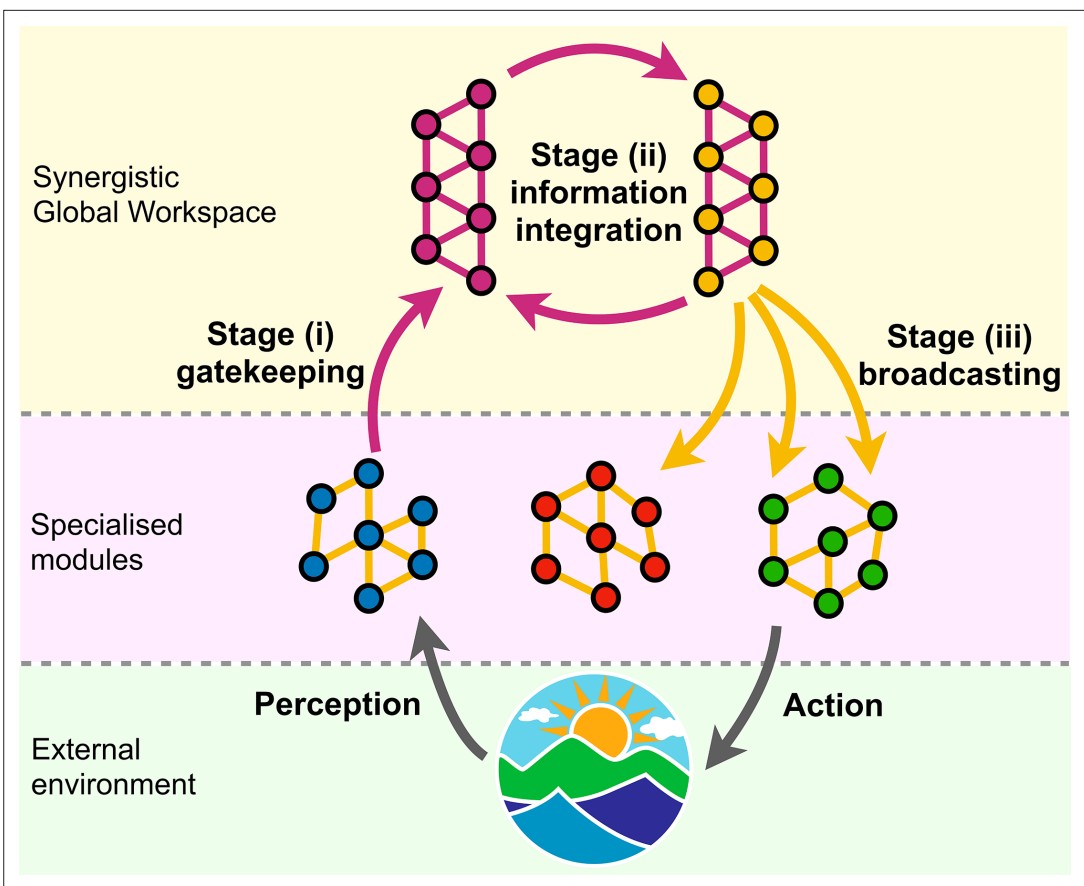

**Figure 1.** Schematic of the proposed SAPHIRE neurocognitive architecture. Below, specialised modules characterised by robust redundant functional interactions process information about the environment. Information is then collected by workspace gateways through synergistic interactions [Stage (i)]; synergistic interactions integrate information within the synergistic global workspace [Stage (ii)]; workspace broadcasters spread the integrated information back to the specialised modules, through redundant interactions [Stage (iii)], for further processing and to guide behaviour. Orange links represent redundant interactions, and violet links represent synergistic interactions. Grey arrows represent interactions between the system and its environment.

## Results

Adopting an information-resolved view, we propose to divide the information-processing stream within the human brain in three key stages: (i) gathering of information from multiple distinct modules into a workspace; (ii) integration of the gathered information within the workspace; and (iii) global information broadcasting to the rest of the brain. Furthermore, we propose that while all workspace regions are involved in stage (ii), they are differentially involved in stages (i) and (iii).

The existence of a synergistic workspace and these three processing stages can be seen as emerging from a trade-off between performance and robustness that is inherent to distributed systems. Theoretical work in cognitive science (*Baars, 2005*) and the field of distributed signal processing (*Tsitsiklis, 1989*; *Veeravalli and Varshney, 2012*) has long recognised the computational benefits of combining multiple distinct processing streams. However, having a single source of inputs to and outputs from the workspace introduces what is known as a 'single point of failure,' which can lead to catastrophic failure in case of damage or malfunction (*Lever et al., 2013*). Therefore, a natural solution is to have not a single but multiple units dedicated to gathering and broadcasting information, respectively, thereby forming a workspace that can be in charge of synthesising the results of peripheral processing (*Rosas et al., 2017*).

Pertaining to Stage (ii), we previously identified which regions of the human brain predominantly entertain synergistic interactions, and thus are most reliant on combining information from other brain regions (*Luppi et al., 2022b*; *Figure 2—figure supplement 1*). The key signature of workspace

regions is to have a high prevalence of synergistic (compared to redundant) functional interactions, and therefore the synergy-rich regions that we discovered are ideally poised as GNW candidates. Here, we consider the architecture of the global workspace more broadly, and combine Integrated Information Decomposition with graph-theoretical principles to bring insights about processing stages (i) and (iii) (*Figure 1*). We term this proposal the 'Synergy-Φ-Redundancy' neurocognitive architecture (SAPHIRE) (*Figure 1*).

We note that brain regions through which information gains access to the workspace should exhibit synergistic functional interactions that are widely distributed across the brain, as – by definition – the workspace gathers and synthesises information from a multiplicity of diverse brain modules. Thus, we postulate that regions that mediate the access to the synergistic workspace are functionally connected with multiple modules within networks of synergistic interactions, synthesising incoming inputs from diverse sources (*Sneve et al., 2019*; *Shanahan, 2012*). We refer to such regions as *gateways* (*Figure 1*, violet nodes). In contrast, the process of broadcasting information corresponds to disseminating multiple copies of the same information from the workspace to many functionally adjacent brain regions. Therefore, broadcaster regions also have functional interactions with many different modules, but of non-synergistic, redundant interactions: 'redundancy' accounts for the fact that multiple copies of the same information are being distributed. These regions are designated as *broadcasters* (*Figure 1*, orange nodes).

One approach to operationalise these ideas is by leveraging well-established graph-theoretical tools. Here, we propose to assess the diversity of intermodular functional connections using the *participation coefficient* (*Rubinov and Sporns, 2010*) which captures to what extent a given node connects to many modules beyond its own (Materials and methods). Note that this is different from the node *strength*, which captures a region's total amount of connectivity, and which we used to identify which regions belong to the synergistic workspace (see Materials and methods and *Luppi et al., 2022b*); the participation coefficient instead quantifies the diversity of modules that a region is connected to. Therefore, gateways are identified from rs-fMRI data as brain regions that (a) belong to the workspace (i.e. have high total synergy), and (b) have a highly ranked participation coefficient in terms of synergistic functional interactions. Conversely, broadcasters are global workspace regions (i.e. also having high synergy) that have a highly ranked participation coefficient rank for redundant interactions.

In other words, we identify the synergistic workspace as regions where synergy predominates, which as our previous research has shown, are also involved with high-level cognitive functions and anatomically coincide with transmodal association cortices at the confluence of multiple information streams (*Luppi et al., 2022b*). This is what we should expect of a global workspace. Subsequently, to discern broadcasters from gateways within the synergistic workspace, we seek to encapsulate the meaning of a 'broadcaster' in information terms. We argue that this corresponds with making the same information available to multiple modules. Sameness of information corresponds to redundancy, and connection with multiple modules can be reflected in the network-theoretic notion of participation coefficient. Thus, a broadcaster is a region in the synergistic workspace (i.e. a region with strong synergistic interactions) that in addition has a high participation coefficient for its redundant interactions.

To explore these hypotheses, we quantified synergistic and redundant interactions between 454 cortical and subcortical brain regions (*Luppi and Stamatakis, 2021*; *Schaefer et al., 2018*) based on resting-state functional MRI data from 100 subjects of the Human Connectome Project (*Luppi et al., 2022b*). Specifically, we systematically applied Integrated Information Decomposition to groups of four variables: the past and future of region X, and the past and future of region Y, for all combinations of X and Y. This provided us with a full decomposition of how information is jointly conveyed by X and Y (redundantly, uniquely, or synergistically) across time. In particular, following our previous work (*Luppi et al., 2022b*) we focused on the persistent synergy (henceforth simply synergy) and persistent redundancy (henceforth simply redundancy), which correspond to the information that is always carried synergistically (respectively, redundantly) by X and Y.

We then subdivided the brain into the well-established resting-state networks identified by Yeo and colleagues (*Yeo et al., 2011*) plus an additional subcortical module (*Tian et al., 2020*). Based on this partition into modules, we identified gateways and broadcasters by comparing the participation coefficients of synergistic versus redundant interactions, for brain regions belonging to the synergistic workspace previously identified we show a significant correlation for participation coefficient

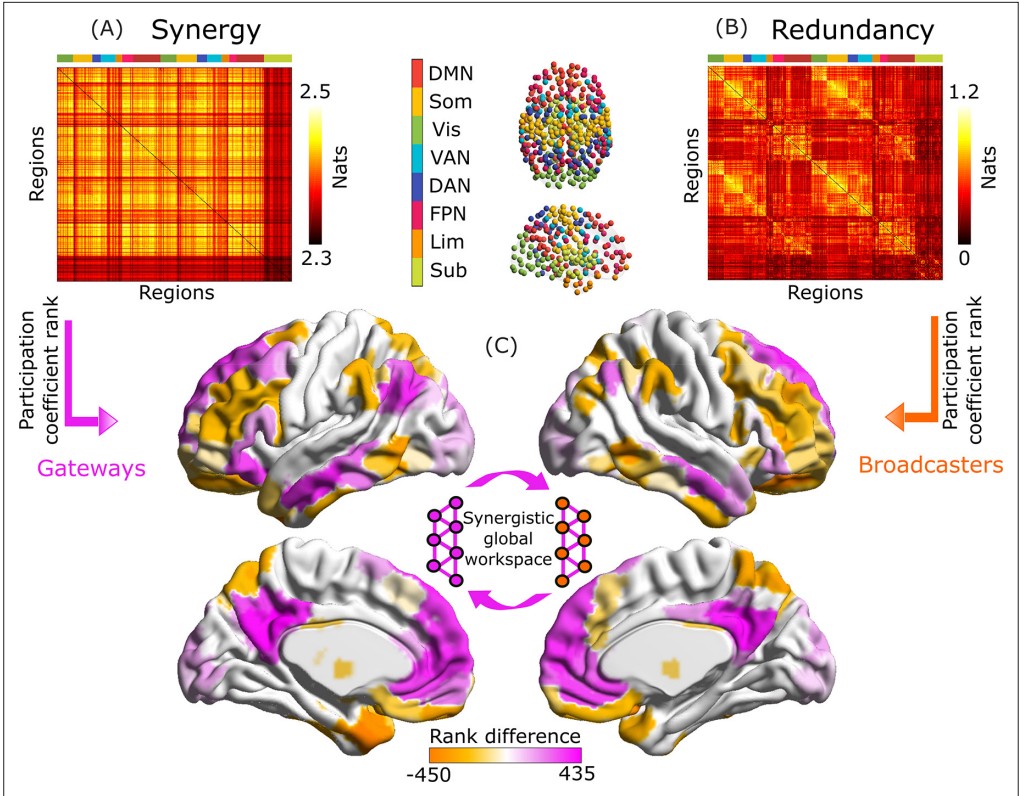

**Figure 2.** Gateways and broadcaster regions identified by their network connectivity profiles. (**A**) Group-average matrix of synergistic interactions between regions of the 454-ROI augmented Schaefer atlas. (**B**) Group-average matrix of redundant interactions. For ease of visualization, the colorscale in (**B**) pertains to log-transformed values. We highlighted modular allegiance to the canonical resting-state networks by using the colour scheme shown in between A and B. (**C**) Regions are identified as gateways (violet) or broadcasters (orange) based on the difference between rank of participation coefficient for synergy and redundancy, (only shown for brain regions identified as belonging to the synergistic global workspace, as per *Luppi et al., 2022b*). Violet indicates synergy rank >redundancy rank, corresponding to workspace regions that combine information of many brain modules (gateways); orange indicates the opposite, identifying workspace regions that broadcast information to many modules. Inset: illustration of the synergistic workspace. Legend: DMN, default mode network. Som, somatomotor network. Vis, visual network. VAN, ventral attention network. DAN, dorsal attention network. FPN, fronto-parietal control network. Lim, limbic network. Sub, subcortical network (comprised of 54 regions of the atlas of *Tian et al., 2020*). These results were also replicated using an alternative parcellation with 232 cortical and subcortical nodes (*Figure 2—figure supplement 3*).

The online version of this article includes the following source data and figure supplement(s) for figure 2:

**Source data 1.** Source data associated with *Figure 2*.

**Figure supplement 1.** Identification of the synergistic workspace.

**Figure supplement 2.** Identification of workspace gateways and broadcasters is robust to node definition.

**Figure supplement 3.** Significant correlation between regional participation coefficient computed with modules defined as resting-state networks (X-axis), and with modules defined from Louvain modularity detection (Y-axis).

obtained from modules defined a priori as the well-known resting-state networks, or defined in a data-driven fashion from Louvain community detection (*Blondel et al., 2008*; *Figure 2—figure supplement 2*).

Intriguingly, our results reveal that gateways reside primarily in the brain's default mode network (*Figure 2B*, violet). In contrast, broadcasters are mainly located in the executive control network, especially lateral prefrontal cortex (*Figure 2B*, orange). Remarkably, the latter results are in line with Global Neuronal Workspace Theory, which consistently identifies lateral prefrontal cortex as a major broadcaster of information (*Mashour et al., 2020*; *Bor and Seth, 2012*).

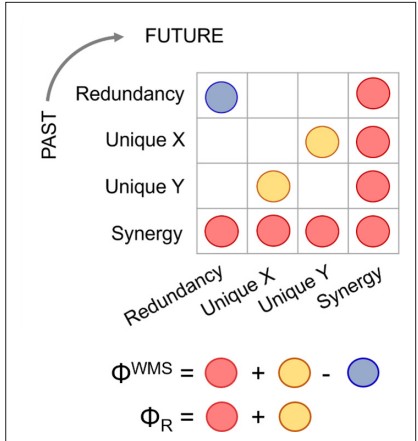

**Figure 3.** Integrated Information Decomposition. Integrated Information Decomposition identifies how two sources X and Y jointly convey information across time, corresponding to all possible 4x4 combinations of redundancy, unique information (of X and of Y), and synergistic information. This decomposition highlights why the original whole-minus-sum $\Phi$ measure introduced by Balduzzi and Tononi can be negative: because it involves the subtraction of the persistent redundancy that is present in the system, leading to negative values in systems that are redundancy-dominated. This shortcoming can be corrected with the revised measure of $\Phi$, termed $\Phi_R$.

## Information decomposition identifies a synergistic core supporting human consciousness

Having introduced a taxonomy within the synergistic global workspace based on the distinct informational roles of different brain regions, we then sought to investigate their role in supporting human consciousness. Given the importance attributed to integration of information by both GNWT and IIT, we expected to observe reductions in integrated information within the areas of the synergistic workspace associated with loss of consciousness. Furthermore, we also reasoned that any brain regions that are specifically involved in supporting consciousness should 'track' the presence of consciousness: the reductions should occur regardless of how loss of consciousness came about, and they should be restored when consciousness is regained.

We tested these hypotheses with resting-state fMRI from 15 healthy volunteers who were scanned before, during, and after anaesthesia with the intravenous agent propofol, as well as 22 patients with chronic disorders of consciousness (DOC) (*Luppi et al., 2019*). Resting-state fMRI data were parcellated into 400 cortical regions from the Schaefer atlas, and 54 subcortical brain regions from the Tian atlas (same parcellation as for the previous analysis). Building on the IIT literature, which provides a formal definition of integrated information, we assessed integration corresponding to conscious activity via two alternative metrics: the well-known whole-minus-sum $\Phi$ measure introduced by *Balduzzi and Tononi, 2008*, and the 'revised $\Phi$' ($\Phi_R$) measure recently introduced by Mediano, Rosas and colleagues (*Mediano et al., 2021*) (Materials and methods and *Figure 3*). Being demonstrably non-negative, this revised measure overcomes a major conceptual limitation of the original formulation of integrated information (*Mediano et al., 2021*).

For each subject, we computed the integrated information between each pair of BOLD signal timeseries, resulting in a 454-by-454 matrix of integrated information between brain regions. Treating this matrix as an (undirected) network enabled us to study consciousness-related changes in integrated information across conditions, which were analysed using the Network Based Statistic correction for multiple comparisons (*Zalesky et al., 2010*). Importantly, since we are interested in changes that are shared between the DOC and propofol datasets, we computed edge-level statistics using a composite null hypothesis test designed to detect such shared effects (Materials and methods).

Analysis based on $\Phi_R$ revealed a widespread reorganisation of integrated information throughout the brain when comparing awake volunteers against DOC patients, with both increases and decreases being observed (p<0.001; *Figure 4A*). Likewise, propofol anaesthesia was also characterised by significant changes in integrated information between brain regions, both when compared with pre-anaesthetic wakefulness (p<0.001; *Figure 4B*) and post-anaesthetic recovery (p<0.001; *Figure 4C*).

Our analysis identified a number of the $\Phi_R$ connections that were reduced when consciousness was lost due to both anaesthesia and brain injury, and were restored during post-anaesthetic recovery – as we had hypothesised (*Figure 4D*). Remarkably, almost all regions showing consistent decreases in $\Phi_R$ when consciousness was lost were members of the global synergistic workspace, and specifically located in the default mode network (bilateral precuneus and medial prefrontal cortex) – and bilateral inferior parietal cortex – although left temporal cortices were also involved (*Figure 4D*). Additionally, some connections exhibited increases in $\Phi_R$ during loss of consciousness, and were restored upon recovery (*Figure 4D*), including areas in frontal cortex – especially lateral prefrontal cortex.

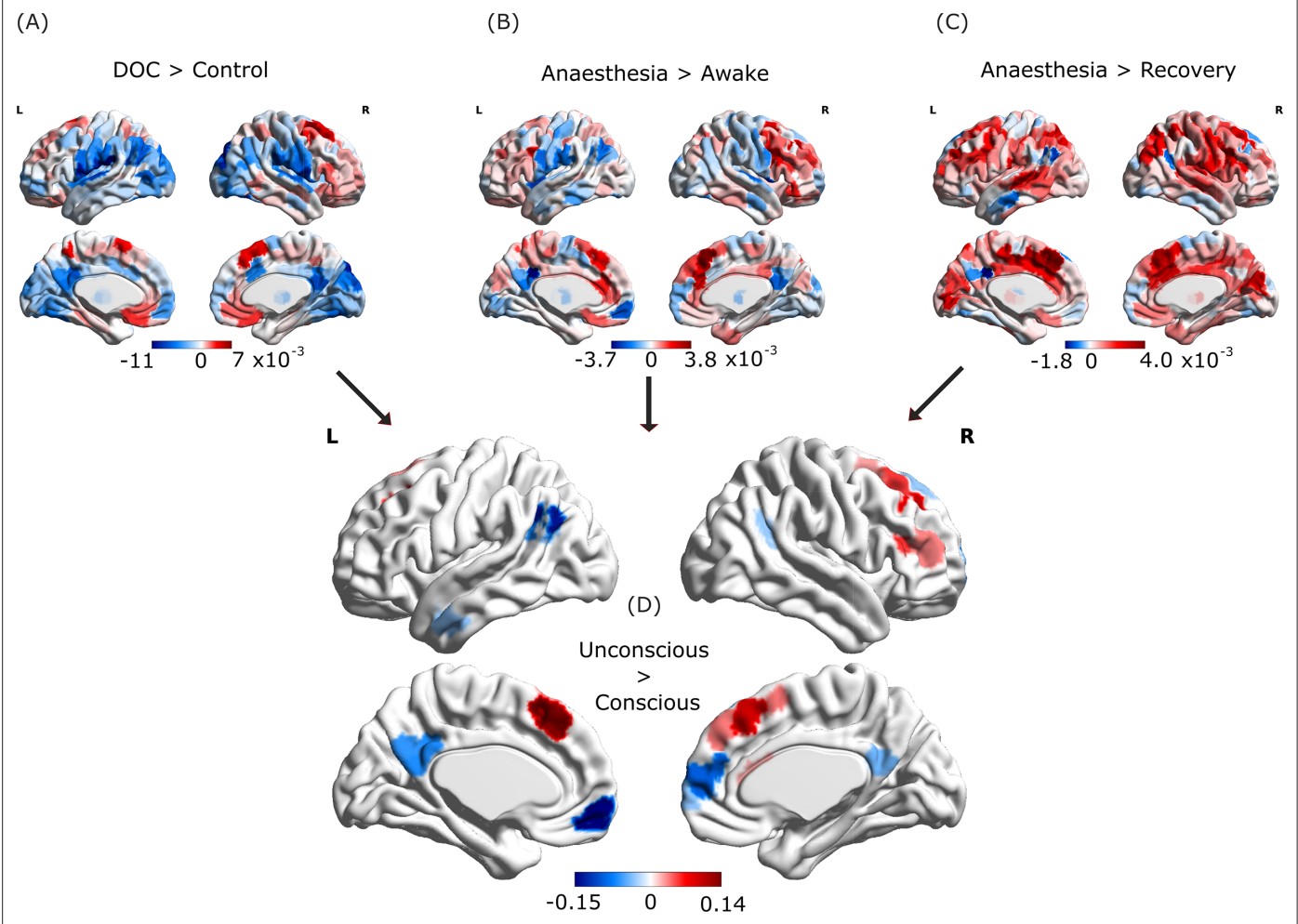

**Figure 4.** Loss of consciousness induces similar reorganisation of cortical integrated information across anaesthesia and disorders of consciousness. Top: Brain regions exhibiting overall NBS-corrected increases (red) and decreases (blue) in integrated information exchange when consciousness is lost. (**A**) DOC patients minus awake healthy volunteers; (**B**), propofol anaesthesia minus pre-induction wakefulness; (**C**) propofol-anaesthesia minus post-anaesthetic recovery. (**D**) Overlaps between the three contrasts in (**A–C**), showing increases and decreases that are common across anaesthesia and disorders of consciousness.

The online version of this article includes the following source data and figure supplement(s) for figure 4:

**Source data 1.** Source data associated with *Figure 4* and *Figure 4—figure supplement 2*.

**Figure supplement 1.** Histogram of significant connectivity changes.

**Figure supplement 2.** Results of alternative analysis choices.

Nevertheless, the overall balance was in favour of reduced integrated information: sum of F-scores associated with significant edges = –25.37 (*Figure 4—figure supplement 1*).

These results were in contrast with the analysis based on the original formulation of $\Phi$ introduced by *Balduzzi and Tononi, 2008*, which did not identify any reductions in integrated information that were common across anaesthesia and disorders of consciousness, instead only identifying common increases (*Figure 4—figure supplement 2*).

Having identified the subset of brain regions that are reliably associated with supporting human consciousness in terms of their integrated information, the last step of our analysis was to leverage the architecture proposed above to understand their role in our information-based view of the global workspace. Since IIT predicts that loss of consciousness corresponds to reductions in integrated information, we focused on regions exhibiting reliable reductions in $\Phi_R$ when consciousness is lost (whether due to anaesthesia or DOC), which were restored upon recovery (shown in blue in *Figure 4D*).

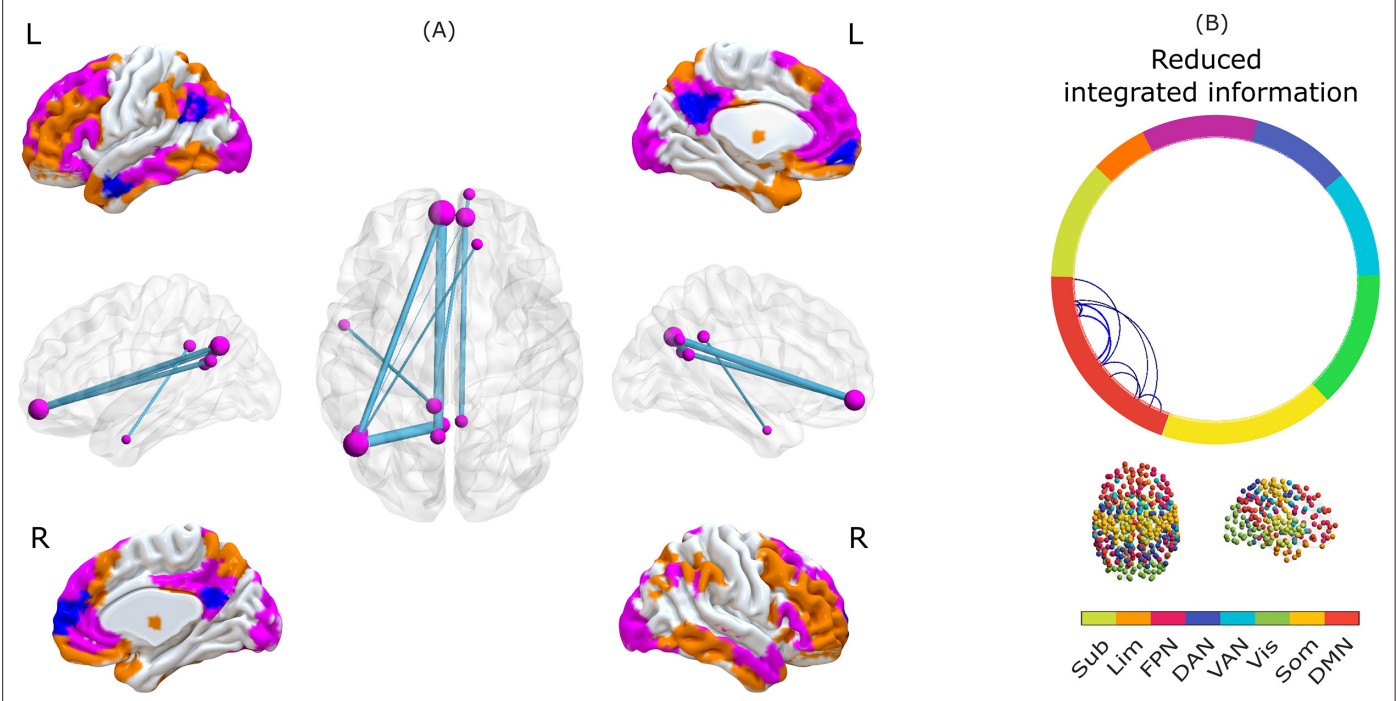

**Figure 5.** Synergistic core of human consciousness. (**A**) Surface representations of medial and lateral views of the brain (L indicates left, R indicates right). Colours indicate brain regions that belong to the synergistic workspace, as identified from HCP data. Orange indicates broadcasters, and violet indicates gateways. Blue indicates regions that exhibit an overall significant reduction in integrated information across anaesthesia and disorders of consciousness. All blue regions overlap with violet ones. The network representation indicates edges with significantly reduced integrated information ($\Phi_R$) during both propofol anaesthesia and disorders of consciousness. The color of the nodes (violet or orange) indicates whether the corresponding regions are workspace gateways (violet) or broadcasters (orange); all regions are gateways (violet). (**B**) Circular graph representation of significant reductions in integrated information ($\Phi_R$) between brain regions, observed in all three contrasts, indicating membership of canonical resting-state networks. Connections indicate pairs of regions with a significant decrease of integrated information. Colour of the circle border indicates the RSN affiliation of the corresponding regions. Legend: DMN, default mode network. Som, somatomotor network. Vis, visual network. VAN, ventral attention network. DAN, dorsal attention network. FPN, fronto-parietal control network. Lim, limbic network. Sub, subcortical network (comprised of 54 regions of the *Tian et al., 2020* atlas).

The online version of this article includes the following source data for figure 5:

**Source data 1.** Source data associated with *Figure 5*.

Remarkably, our whole-brain results show that $\Phi_R$ disconnections induced by loss of consciousness play the role of gateway nodes (*Figure 5A*, violet) rather than broadcaster nodes (*Figure 5A*, orange) according to our previous identification of gateways and broadcasters from the Human Connectome Project dataset (see *Figure 2B*, violet regions). Indeed, all reductions occur specifically within the default mode network (*Figure 5B*). Thus, these results suggest that loss of consciousness across anaesthesia and disorders of consciousness would correspond to anterior-posterior disconnection – in terms of integrated information – between DMN nodes that act as gateways into the synergistic workspace.

## Robustness and sensitivity analysis

To ensure the robustness of our results to analytic choices, we also replicated them using an alternative cortical parcellation of lower dimensionality: we used the Schaefer scale-200 cortical parcellation (*Schaefer et al., 2018*) complemented with the scale-32 subcortical ROIs from the Tian subcortical atlas (*Tian et al., 2020Figure 4—figure supplement 2*). Additionally, we also show that our results are not dependent on the choice of parameters in the NBS analysis, and are replicated using an alternative threshold definition for the connected component (extent rather than intensity) or a more stringent value for the cluster threshold (*F*>12; *Figure 4—figure supplement 2*). Importantly, whereas the increases in $\Phi_R$ are not the same across different analytic approaches, reductions of $\Phi_R$ in medial prefrontal and posterior cingulate/precuneus are reliably observed, attesting to their robustness.

## Discussion

### Architecture of the synergistic global workspace

This paper proposes an informational perspective on the brain's functional architecture at the macroscale, which leverages insights from network science and a refined understanding of neural information exchange. The synergy-$\Phi$-redundancy (SAPHIRE) architecture posits the existence of a 'synergistic workspace' of brain regions characterised by highly synergistic global interactions, which we previously showed to be composed by prefrontal and parietal cortices that are critical for higher cognitive functions (*Luppi et al., 2022b*). This workspace is further functionally decomposed by distinguishing gateways, which bring information from localised modules into the workspace, and broadcasters, which disseminate multiple copies of workspace information back to low-level regions.

Remarkably, our results on the HCP dataset show that the proposed operationalisation of gateways and broadcasters corresponds to the distinction between the brain's default mode network and executive control network, respectively. This data-driven identification of workspace gateways and broadcasters with the DMN and FPN provides a new framework to explain well-known functional differences between DMN and FPN, based on their distinct and complementary roles within the brain's synergistic global workspace, which is discussed below.

The fronto-parietal executive control network (FPN) mainly comprises lateral prefrontal and parietal cortices, and it is associated with performance of a variety of complex, cognitively demanding tasks (*Fedorenko et al., 2013*; *Duncan and Owen, 2000*; *Barbey, 2018*). A key component of this network is lateral prefrontal cortex (LPFC). Based on theoretical and empirical evidence, as summarised in a recent review of GNWT (*Mashour et al., 2020*), this region is posited to play a major role in the global workspace, as a global broadcaster of information. Remarkably, this is precisely the role that our results assigned to LPFC, based on its combined information-theoretic and network properties. These results are also consistent with recent insights from network neuroscience, which indicate that the FPN is ideally poised to steer whole-brain dynamics through novel trajectories, in response to complex task demands (*Barbey, 2018*; *Gu et al., 2015*). Specifically, by broadcasting to the rest of the brain information that has been integrated within the workspace, the FPN may act as global coordinator of subsequent whole-brain dynamics.

On the other hand, the default mode network comprises posterior cingulate and precuneus, medial prefrontal cortex, and inferior parietal cortices (*Fox et al., 2005*; *Raichle et al., 2001*; *Raichle, 2015*). This network, whose constituent regions have undergone substantial developments in the course of human evolution (*Wei et al., 2019*; *Xu et al., 2020*), was found to occupy a crucial position at the convergence of functional gradients of macroscale cortical organization (*Margulies et al., 2016*; *Huntenburg et al., 2018*; *Smallwood et al., 2021*), forming a structural and functional core of the human brain (*Deco et al., 2017*; *Kabbara et al., 2017*; *de Pasquale et al., 2012*), in line with its recently observed involvement in cognitive tasks (*Vatansever et al., 2015*; *Vatansever et al., 2017*; *Chiou et al., 2020*). In particular, the DMN is prominently involved in self-referential processing (*Cavanna and Trimble, 2006*; *Qin and Northoff, 2011*), and 'mental-time-travel' (*Karapanagiotidis et al., 2017*) or episodic memory and future-oriented cognition (*Buckner et al., 2008*; *Buckner and DiNicola, 2019*; *Schacter et al., 2007*; *Szpunar et al., 2014*). Its posterior regions in particular, act as relays between the neocortex and the hippocampal memory system (*Buckner and DiNicola, 2019*). Thus, in terms of both neuroanatomical connectivity and functional engagement, the DMN is uniquely positioned to integrate and contextualise information coming into the synergistic global workspace (e.g. from sensory streams) by combining it with rich information pertaining to one's past experiences and high-level mental models about 'self' and world (*Smallwood et al., 2021*; *Hassabis and Maguire, 2009*; *Wen et al., 2020*; *Wang et al., 2020*; *Dohmatob et al., 2020*; *Yeshurun et al., 2021*) – coinciding with the results of the present analysis, which identify DMN nodes as gateways of inputs to the synergistic global workspace.

It is worth noting that the role of the FPN-DMN tandem in supporting consciousness has been suggested by Shanahan's hypothesis of a 'connective core' along the brain's medial axis (*Shanahan, 2010*). While Shanahan's hypotheses were primarily based on structure, in this work we combine novel information-theoretic tools to confirm and expand the connective core hypothesis from a functional, information-centric perspective, in a way that differentiates the multiple roles played by the different regions that together comprise this connective core (*Figures 2 and 5*).

## Integrated information decomposition of human consciousness

After identifying the neuroanatomical-functional mapping of the synergistic workspace in terms of gateways and broadcasters, we sought to identify their role in supporting human consciousness. Considering integrated information as a marker of consciousness (without necessarily assuming the two to be identical), we focused on identifying regions where information integration is reduced when consciousness is lost (regardless of its cause, be it propofol anaesthesia or severe brain injury), and restored upon its recovery. Our results indicate that brain regions exhibiting consciousness-specific reductions in integrated information coincide with major nodes of the synergistic global workspace.

Intriguingly, we found that the main disruptions of information integration were localised in gateway nodes, rather than broadcasters. Thus, loss of consciousness in both anaesthesia and disorders of consciousness could be understood as a breakdown of the entry points to the 'synergistic core' (*Figure 5*), which becomes unable to properly integrate inputs for the workspace. Importantly, the original 'whole-minus-sum' $\Phi$ introduced by *Balduzzi and Tononi, 2008* did not show consistent reductions during loss of consciousness. Thus, the present results demonstrate the empirical validity of the 'revised' measure, $\Phi_R$, in addition to its theoretical soundness (*Mediano et al., 2021*). Since workspace gateway regions coincide with the brain's default mode network, these results are also in line with recent evidence that information content and integrative capacity of the DMN are compromised during loss of consciousness induced by both anaesthesia and severe brain injury (*Luppi et al., 2019*; *MacDonald et al., 2015*; *Hannawi et al., 2015*; *Vanhaudenhuyse et al., 2010*; *Di Perri et al., 2018*; *Demertzi et al., 2015*; *Boveroux et al., 2010*; *Bodien et al., 2019*; *Spindler et al., 2021*; *Huang et al., 2020*), and even COVID-19 (*Fischer et al., 2022*). Due to its prominent role in self-referential processing (*Qin and Northoff, 2011*), breakdown of DMN connectivity within the synergistic workspace may be seen as a failure to integrate one's self-narrative into the 'stream of consciousness', in the words of William James.

This notion is further supported by focusing on reductions of integrated information during anaesthesia compared with wakefulness. In addition to the synergistic core, overall reductions are also observed in a set of thalamic, auditory and somatomotor regions, largely resembling the brain regions that stop responding to sensory (auditory and noxious) stimuli once the brain reaches propofol-induced saturation of EEG slow-wave activity (SWAS *Ní Mhuircheartaigh et al., 2013*). Although there was no EEG data available to confirm this, the doses of propofol employed in the present study are compatible with the doses of propofol at which SWAS has been shown to arise (*Warnaby et al., 2017*), and therefore it is plausible that our participants also reached SWAS and the loss of brain responsiveness it indicates. Thus, both resting-state integration of information between brain regions, as well as stimulus-evoked responses within each region (*Ní Mhuircheartaigh et al., 2013*), converge to indicate that propofol disrupts further processing of thalamocortical sensory information – a phenomenon termed 'thalamocortical isolation' (*Ní Mhuircheartaigh et al., 2013*). We propose that as the thalamus and sensory cortices lose their ability to respond to stimuli, they cease to provide information to the synergistic core of the global workspace, resulting in a disconnection from the external world and presumably loss of consciousness.

These results testify to the power of the Integrated Information Decomposition framework: by identifying the information-theoretic components of integrated information, we have been able to obtain insights about human consciousness that remained elusive with alternative formulations, and could not be captured via standard functional connectivity or related methods. Thus, our findings are consistent with the notion that the global workspace is relevant for supporting consciousness in the human brain, in line with the proposal that '[...] unconsciousness is not necessarily a complete suppression of information processing but rather a network dysfunction that could create inhospitable conditions for global information exchange and broadcasting' (*Mashour et al., 2020*). GNWT postulates a key role for the global workspace in supporting consciousness: consistent with this theory, we find that several nodes of the synergistic global workspace become disconnected from each other in terms of integrated information when consciousness is lost, especially between anterior and posterior regions (*Figure 4*, brain networks). Thus, these are brain regions that (i) belong to the synergistic global workspace; (ii) exhibit overall reductions of integrated information when consciousness is lost; and (iii) are disconnected from other regions of the synergistic workspace when consciousness is lost. The brain regions satisfying these three conditions therefore meet the criteria for constituting an interconnected 'synergistic core' of workspace regions supporting human consciousness.

## Limitations and future directions

In order to obtain high spatial resolution for our identification of workspace regions, here we relied on the BOLD signal from functional MRI, which is an indirect proxy of underlying neuronal activity, with limited temporal resolution. However, we sought to alleviate potential confounds by deconvolving the hemodynamic response function from our data with a dedicated toolbox (*Wu et al., 2013*; Materials and methods), which has been previously applied both in the context of information decomposition (*Luppi et al., 2022b*), as well as anaesthetic-induced loss of consciousness (*Wu et al., 2019*), and disorders of consciousness (*Luppi et al., 2023a*). Additionally, the present results of an overall $\Phi_R$ reduction are also broadly in line with those of a previous study (*Faes et al., 2022*), whose measure of synergy-redundancy balance showed, in ECoG recordings of non-human primates, a broadband shift away from synergy during anaesthesia.

It is also worth bearing in mind that our measure of integrated information between pairs of regions does not amount to measuring the integrated information of the brain as a whole, as formally specified in the context of Integrated Information Theory (*Balduzzi and Tononi, 2008*) - although we do show that the average integrated information between pairs of regions is overall reduced across the whole brain. We also note that our revised measure of integrated information is based on IIT 2.0 (*Balduzzi and Tononi, 2008*), due to its computational tractability; as a result, it relies on a conceptually distinct understanding of integrated information from the more recent IIT 3.0 (*Oizumi et al., 2014*) and IIT 4.0 (*Albantakis et al., 2023*) versions. Thus, these limitations should be borne in mind when seeking to interpret the present results in the context of IIT. Indeed, future work may benefit from seeking convergence with recent advances in the characterization of emergence, which is related to integrated information (*Hoel et al., 2013*; *Hoel et al., 2016*; *Klein and Hoel, 2020*; *Varley and Hoel, 2022*).

Likewise, it is not our intention to claim that $\Phi_R$ and synergy, as measured at the level of regional BOLD signals, represent a direct cause of consciousness, or are identical to it. Rather, our work is intended to use these measures similarly to the use of sample entropy and Lempel-Ziv complexity for BOLD signals (*Luppi et al., 2019*; *Varley et al., 2020b*): as theoretically grounded macroscale indicators, whose empirical relationship to consciousness may point towards the relevant underlying neural phenomena. In other words, while our results do show that BOLD-derived $\Phi_R$ tracks the loss and recovery of consciousness, we do not claim that they are the cause of it: only that an empirical relationship exists, which is in line with what we might expect on theoretical grounds. Future work will be required to identify whether this empirical relationship also holds at the microscale, and whether the causal mechanisms that induce loss of consciousness are also causally responsible for loss of integrated information.

Intriguingly, although we have focused on anaesthetic-induced decreases in integrated information, due to IIT's prediction that this is what should occur during loss of consciousness, our results also indicate concomitant increases of integrated information – possibly reflecting compensatory attempts, although we refrain from further speculation (*Figure 4*). Interestingly, increases appear to coincide with broadcaster nodes of the synergistic workspace. In particular, even though lateral prefrontal cortices are among the regions most closely associated with the global neuronal workspace in the literature (*Mashour et al., 2020*; *Bor and Seth, 2012*), our results indicate a paradoxical net increase in lateral prefrontal integrated information during anaesthesia and DOC. We interpret this qualitatively different behaviour as indicating that different subsets of the global workspace may be differentially involved in supporting consciousness.

However, we note that, whereas the decreases in integrated information were robust to the use of different analytic approaches (e.g., use of a different parcellation or different NBS threshold), the increases that we observed were less robust, with no region consistently showing increases in integrated information (*Figure 4—figure supplement 2*). Nevertheless, both this phenomenon and the meaning of increased integrated information between brain regions deserve further investigation. Indeed, dreaming during anaesthesia has been reported to occur in up to 27% of cases (*Leslie et al., 2007*), and behaviourally unresponsive participants have been shown to perform mental imagery tasks during anaesthesia, both of which constitute cases of disconnected consciousness (*Huang et al., 2018*). Thus, although our doses of propofol were consistent with the presence of SWAS, we cannot exclude that some of our participants may have been merely disconnected but still conscious, possibly driving the increases we observed.

More broadly, future research may also benefit from characterising the role of the synergistic workspace in the states of altered consciousness induced e.g. by psychedelics (*Luppi et al., 2023c*; *Huang et al., 2023*; *Timmermann et al., 2023*), especially since prominent involvement of the DMN has already been identified (*Carhart-Harris, 2018*; *Carhart-Harris et al., 2016*). Likewise, the use of paradigms different from resting-state, such as measuring the brain's spontaneous responses to engaging stimuli (e.g. suspenseful narratives *Naci et al., 2017*) or engaging movies (*Naci et al., 2014*) may provide evidence for a more comprehensive understanding of brain changes during unconsciousness. Likewise, it will be of great interest to investigate whether and how reorganization of the synergistic global workspace is reflected in other indicators of consciousness (*Lee et al., 2022*), such as the brain's response to external perturbations – such as the EEG response to brief magnetic pulses used to compute the Perturbational Complexity Index, one of the most discriminative indices of consciousness available to date (*Ferrarelli et al., 2010*; *Bodart et al., 2017*; *Casali et al., 2013*; *Casarotto et al., 2016*; *Sarasso et al., 2015*).

The PCI is used as a means of assessing the brain's current state, but stimulation protocols can also be adopted to directly induce transitions between states of consciousness. In rodents, carbachol administration to frontal cortex awakens rats from sevoflurane anaesthesia (*Pal et al., 2018*), and optogenetic stimulation was used to identify a role of central thalamus neurons in controlling transitions between states of responsiveness (*Liu et al., 2015*; *Gent et al., 2018*). Additionally, several studies in non-human primates have now shown that electrical stimulation of the central thalamus can reliably induce awakening from anaesthesia, accompanied by the reversal of electrophysiological and fMRI markers of anaesthesia (*Bastos et al., 2021*; *Redinbaugh et al., 2020*; *Tasserie et al., 2022*; *Luppi et al., 2024b*; *Redinbaugh et al., 2022*; *Afrasiabi et al., 2021*). Finally, in human patients suffering from disorders of consciousness, stimulation of intra-lami9nar central thalamic nuclei was reported to induce behavioural improvement (*Schiff, 2008*), and ultrasonic stimulation (*Cain et al., 2021*; *Cain et al., 2022*) and deep-brain stimulation are among potential therapies being considered for DOC patients (*Edlow et al., 2021a*; *Edlow et al., 2021b*). It will be of considerable interest to determine whether our corrected measure of integrated information and topography of the synergistic workspace also restored by these causal interventions.

Additionally, the reliance here on 'resting-state' data without external stimuli may have resulted in an overestimation of the DMN's role in consciousness, and an under-estimation of the FPN (including lateral PFC), given their known different recruitment during no-task conditions (*Fox et al., 2005*). Indeed, recent efforts have been carried out to obtain a data-driven characterisation of the brain's global workspace based on regions' involvement across multiple different tasks (*Deco et al., 2021b*). This work is complementary to ours in two aspects: first, the focus of *Deco et al., 2021b* is on the role of the workspace related to cognition, whereas here we focus primarily on consciousness. Second, by using transfer entropy (*Schreiber, 2000*; *Massey, 1990*) as a measure of functional connectivity, Deco and colleagues (*Deco et al., 2021b*) assessed the directionality of information exchange – whereas our measure of integrated information is undirected, but are able to distinguish between different kinds of information being exchanged and integrated. Thus, different ways of defining and characterising a global workspace in the human brain are possible, and can provide complementary insights about distinct aspects of the human neurocognitive architecture. Indeed, transfer entropy can itself be decomposed into information-dynamic atoms through Partial Information Decomposition and Integrated Information Decomposition (*Williams and Beer, 2010*; *Luppi et al., 2024a*; *Mediano et al., 2021*; *Williams and Beer, 2011*); ΦID can further decompose the Normalised Directed Transfer Entropy measure used by *Deco et al., 2021b*, as recently demonstrated (*Luppi et al., 2023b*). We look forward to a more refined conceptualization of the synergistic workspace architecture that takes into account both information types and the directionality of information flow – especially in datasets with higher temporal resolution.

Looking forward, growing evidence indicates an important role for brain dynamics and time-resolved brain states in supporting cognition (*Shine et al., 2019*; *Deco et al., 2021a*; *Shine et al., 2016*; *Zamani Esfahlani et al., 2020*; *Faskowitz et al., 2020*; *Lurie et al., 2020*; *Cabral et al., 2023*; *Raut et al., 2021*; *Vidaurre et al., 2017*; *Atasoy et al., 2018*) and consciousness (*Luppi et al., 2019*; *Huang et al., 2020*; *Demertzi et al., 2019*; *Gutierrez-Barragan et al., 2022*; *Luppi et al., 2021a*; *Luppi et al., 2021b*; *Lord et al., 2019*; *Barttfeld et al., 2015*; *Uhrig et al., 2018*; *Stevner et al., 2019*). Therefore, time-resolved extensions of our framework, such as developed by *Varley, 2023*,

may shed further light on the dynamics of the synergistic workspace, especially if combined with neuroimaging modalities offering higher temporal resolution, such as magneto- or electroencephalography. More broadly, a key strength of our proposed cognitive architecture is its generality: being entirely grounded in the combination of information theory and network science, it could be applied to shed light on cognition in humans and other organisms (*Bayne et al., 2020*), but also to inspire further development of artificial cognitive systems (*Luppi et al., 2024a*; *Connor and Shanahan, 2007*; *Shanahan, 2006*; *Langdon et al., 2022*; *VanRullen and Kanai, 2021*; *Proca et al., 2024*).

## Conclusion

Overall, we have shown that powerful insights about human consciousness and neurocognitive architecture can be obtained through the information-resolved approach, afforded by the framework of Integrated Information Decomposition. Importantly, the proposed criteria to identify gateways, broadcasters, and the synergistic workspace itself, are based on practical network and information-theoretic tools, which are applicable to a broad range of neuroimaging datasets and neuroscientific questions. These findings bring us closer to a unified theoretical understanding of consciousness and its neuronal underpinnings - how mind arises from matter.

# Materials and methods

**Key resources table**

| Reagent type (species) or resource | Designation | Source or reference | Identifiers | Additional information |
|---|---|---|---|---|
| Software, algorithm | Java Information Dynamics Toolbox | *Lizier, 2014* | https://github.com/jlizier/jidt | v1.5 |
| Software, algorithm | CONN toolbox | *Whitfield-Gabrieli and Nieto-Castanon, 2012* | http://www.nitrc.org/projects/conn | version 17 f |
| Software, algorithm | Brain Connectivity Toolbox | *Rubinov and Sporns, 2010* | https://sites.google.com/site/bctnet/ | |
| Software, algorithm | HRF deconvolution toolbox | *Wu et al., 2013* | https://www.nitrc.org/projects/rshrf | v2.2 |
| Software, algorithm | Spin-test | *Alexander-Bloch et al., 2018* | https://github.com/frantisekvasa/rotate_parcellation | |
| Software, algorithm | Integrated Information Decomposition code | *Luppi et al., 2024a* | https://github.com/Imperial-MIND-lab/integrated-info-decomp | |

The propofol and DOC patient functional data employed in this study have been published before (*Luppi et al., 2019*; *Varley et al., 2020b*; *Luppi et al., 2023c*; *Luppi et al., 2022a*; *Naci et al., 2018*; *Kandeepan et al., 2020*; *Varley et al., 2020a*). For clarity and consistency of reporting, where applicable we use the same wording as our previous work (*Luppi et al., 2019*; *Luppi et al., 2023c*; *Luppi et al., 2022a*).

## Anaesthesia data: Recruitment

The propofol data were collected between May and November 2014 at the Robarts Research Institute in London, Ontario (Canada); (*Luppi et al., 2019*). The study received ethical approval from the Health Sciences Research Ethics Board and Psychology Research Ethics Board of Western University (Ontario, Canada). Healthy volunteers (n=19) were recruited (18–40 years; 13 males). Volunteers were right-handed, native English speakers, and had no history of neurological disorders. In accordance with relevant ethical guidelines, each volunteer provided written informed consent, and received monetary compensation for their time. Due to equipment malfunction or physiological impediments to anaesthesia in the scanner, data from n=3 participants (1 male) were excluded from analyses, leaving a total n=16 for analysis (*Luppi et al., 2019*; *Luppi et al., 2023c*; *Luppi et al., 2022a*).

## Anaesthesia data: Procedure

Resting-state fMRI data were acquired at different propofol levels: no sedation (Awake), and Deep anaesthesia (corresponding to Ramsay score of 5). As previously reported (*Luppi et al., 2019*; *Luppi*

et al., 2023c; Luppi et al., 2022a), for each condition fMRI acquisition began after two anaesthesiologists and one anaesthesia nurse independently assessed Ramsay level in the scanning room. The anaesthesiologists and the anaesthesia nurse could not be blinded to experimental condition, since part of their role involved determining the participants' level of anaesthesia. Note that the Ramsay score is designed for critical care patients, and therefore participants did not receive a score during the Awake condition before propofol administration: rather, they were required to be fully awake, alert and communicating appropriately. To provide a further, independent evaluation of participants' level of responsiveness, they were asked to perform two tasks: a test of verbal memory recall, and a computer-based auditory target-detection task. Wakefulness was also monitored using an infrared camera placed inside the scanner.

Propofol was administered intravenously using an AS50 auto syringe infusion pump (Baxter Healthcare, Singapore); an effect-site/plasma steering algorithm combined with the computer-controlled infusion pump was used to achieve step-wise sedation increments, followed by manual adjustments as required to reach the desired target concentrations of propofol according to the TIVA Trainer (European Society for Intravenous Aneaesthesia, eurosiva.eu) pharmacokinetic simulation program. This software also specified the blood concentrations of propofol, following the Marsh 3-compartment model, which were used as targets for the pharmacokinetic model providing target-controlled infusion. After an initial propofol target effect-site concentration of 0.6 $\mu$g mL$^{-1}$, concentration was gradually increased by increments of 0.3 $\mu$g mL$^{1}$, and Ramsay score was assessed after each increment: a further increment occurred if the Ramsay score was lower than 5. The mean estimated effect-site and plasma propofol concentrations were kept stable by the pharmacokinetic model delivered via the TIVA Trainer infusion pump. Ramsay level 5 was achieved when participants stopped responding to verbal commands, were unable to engage in conversation, and were rousable only to physical stimulation. Once both anaesthesiologists and the anaesthesia nurse all agreed that Ramsay sedation level 5 had been reached, and participants stopped responding to both tasks, data acquisition was initiated. The mean estimated effect-site propofol concentration was 2.48 (1.82–3.14) $\mu$g mL$^{-1}$, and the mean estimated plasma propofol concentration was 2.68 (1.92–3.44) $\mu$g mL$^{-1}$. Mean total mass of propofol administered was 486.58 (373.30–599.86) mg. These values of variability are typical for the pharmacokinetics and pharmacodynamics of propofol. Oxygen was titrated to maintain SpO2 above 96%.

At Ramsay 5 level, participants remained capable of spontaneous cardiovascular function and ventilation. However, the sedation procedure did not take place in a hospital setting; therefore, intubation during scanning could not be used to ensure airway security during scanning. Consequently, although two anaesthesiologists closely monitored each participant, scanner time was minimised to ensure return to normal breathing following deep sedation. No state changes or movement were noted during the deep sedation scanning for any of the participants included in the study (Luppi et al., 2019; Luppi et al., 2023c; Luppi et al., 2022a). Propofol was discontinued following the deep anaesthesia scan, and participants reached level 2 of the Ramsey scale approximately 11 min afterwards, as indicated by clear and rapid responses to verbal commands. This corresponds to the 'recovery' period (Naci et al., 2018).

## Anaesthesia data: Design

As previously reported (Luppi et al., 2019; Luppi et al., 2023c; Luppi et al., 2022a), once in the scanner participants were instructed to relax with closed eyes, without falling asleep. Resting-state functional MRI in the absence of any tasks was acquired for 8 min for each participant, in each condition. A further scan was also acquired during auditory presentation of a plot-driven story through headphones (5 min long). Participants were instructed to listen while keeping their eyes closed. The present analysis focuses on the resting-state data only; the story scan data have been published separately, and will not be discussed further here.

## Anaesthesia data: FMRI data acquisition

As previously reported (Luppi et al., 2019; Luppi et al., 2023c; Luppi et al., 2022a), MRI scanning was performed using a 3-Tesla Siemens Tim Trio scanner (32-channel coil), and 256 functional volumes (echo-planar images, EPI) were collected from each participant, with the following parameters: slices = 33, with 25% inter-slice gap; resolution = 3 mm isotropic; TR = 2000ms; TE = 30ms; flip angle = 75 degrees; matrix size = 64×64. The order of acquisition was interleaved, bottom-up. Anatomical

scanning was also performed, acquiring a high-resolution T1- weighted volume (32-channel coil, 1 mm isotropic voxel size) with a 3D MPRAGE sequence, using the following parameters: TA = 5 min, TE = 4.25ms, 240x256 matrix size, 9 degrees flip angle (*Luppi et al., 2019*; *Luppi et al., 2023c*; *Luppi et al., 2022a*).

## Disorders of consciousness patient data: Recruitment

A total of 71 DOC patients were recruited from specialised long-term care centres from January 2010 to December 2015 (*Luppi et al., 2019*; *Luppi et al., 2023c*; *Luppi et al., 2022a*). Ethical approval for this study was provided by the National Research Ethics Service (National Health Service, UK; LREC reference 99/391). Patients were eligible to be recruited in the study if they had a diagnosis of chronic disorder of consciousness, provided that written informed consent to participation was provided by their legal representative, and provided that the patients could be transported to Addenbrooke's Hospital (Cambridge, UK). The exclusion criteria included any medical condition that made it unsafe for the patient to participate, according to clinical personnel blinded to the specific aims of the study; or any reason that made a patient unsuitable to enter the MRI scanner environment (e.g. non-MRI-safe implants). Patients were also excluded based on substantial pre-existing mental health problems, or insufficient fluency in the English language prior to their injury. After admission to Addenbrooke's Hospital, each patient underwent clinical and neuroimaging testing, spending a total of five days in the hospital (including arrival and departure days). Neuroimaging scanning took place at the Wolfson Brain Imaging Centre (Addenbrooke's Hospital, Cambridge, UK), and medication prescribed to each patient was maintained during scanning.

For each day of admission, Coma Recovery Scale-Revised (CRS-R) assessments were recorded at least daily. Patients whose behavioural responses were not indicative of awareness at any time, were classified as UWS. In contrast, patients were classified as being in a minimally conscious state (MCS) if they provided behavioural evidence of simple automatic motor reactions (e.g. scratching, pulling the bed sheet), visual fixation and pursuit, or localisation to noxious stimulation. Since this study focused on whole-brain properties, coverage of most of the brain was required, and we followed the same criteria as in our previous studies (*Luppi et al., 2019*; *Luppi et al., 2023c*; *Luppi et al., 2022a*); before analysis took place, patients were systematically excluded if an expert neuroanatomist blinded to diagnosis judged that they displayed excessive focal brain damage (over one third of one hemisphere), or if brain damage led to suboptimal segmentation and normalisation, or due to excessive head motion in the MRI scanner (exceeding 3 mm translation or 3 degrees rotation). Of the initial sample of 71 patients who had been recruited, a total of 22 adults (14 males; 17–70 years; mean time post injury: 13 months) meeting diagnostic criteria for Unresponsive Wakefulness Syndrome/Vegetative State or Minimally Conscious State due to brain injury were included in this study. In addition to the researcher and radiographer, a research nurse was also present during scanning. Since the patients' status as DOC patients was evident, no researcher blinding was possible.

## Disorders of consciousness patient data: FMRI data acquisition

As previously reported (*Luppi et al., 2019*; *Luppi et al., 2023c*; *Luppi et al., 2022a*), resting-state fMRI was acquired for 10 min (300 volumes, TR = 2000ms) using a Siemens Trio 3T scanner (Erlangen, Germany). Functional images (32 slices) were acquired using an echo planar sequence, with the following parameters: 3x3 x 3.75mm resolution, TR = 2000ms, TE = 30ms, 78 degrees FA. Anatomical scanning was also performed, acquiring high-resolution T1-weighted images with an MPRAGE sequence, using the following parameters: TR = 2300ms, TE = 2.47ms, 150 slices, resolution 1x1x1mm.

## Functional MRI preprocessing and denoising

The functional imaging data were preprocessed using a standard pipeline, implemented within the SPM12-based (http://www.fil.ion.ucl.ac.uk/spm) toolbox CONN (http://www.nitrc.org/projects/conn), version 17 f (*Whitfield-Gabrieli and Nieto-Castanon, 2012*). The pipeline comprised the following steps: removal of the first five scans, to allow magnetisation to reach steady state; functional realignment and motion correction; slice-timing correction to account for differences in time of acquisition between slices; identification of outlier scans for subsequent regression by means of the quality assurance/artifact rejection software *art* (http://www.nitrc.org/projects/artifact_detect); structure-function coregistration using each volunteer's high-resolution T1-weighted image; spatial normalisation to

Montreal Neurological Institute (MNI-152) standard space with 2 mm isotropic resampling resolution, using the segmented grey matter image, together with an a priori grey matter template.

To reduce noise due to cardiac, breathing, and motion artifacts, which are known to impact functional connectivity and network analyses (*Van Dijk et al., 2012*; *Power et al., 2012*), we applied the anatomical CompCor method of denoising the functional data (*Behzadi et al., 2007*), also implemented within the CONN toolbox. As for preprocessing, we followed the same denoising described in previous work (*Luppi et al., 2019*; *Luppi et al., 2023c*; *Luppi et al., 2022a*). The anatomical CompCor method involves regressing out of the functional data the following confounding effects: the first five principal components attributable to each individual's white matter signal, and the first five components attributable to individual cerebrospinal fluid (CSF) signal; six subject-specific realignment parameters (three translations and three rotations) as well as their first- order temporal derivatives; the artefacts identified by *art*; and main effect of scanning condition *Behzadi et al., 2007*. Linear detrending was also applied, and the subject-specific denoised BOLD signal timeseries were bandpass filtered to eliminate both low-frequency drift effects and high-frequency noise, thus retaining temporal frequencies between 0.008 and 0.09 Hz.

The step of global signal regression (GSR) has received substantial attention in the fMRI literature, as a potential denoising step (*Power et al., 2014*; *Lydon-Staley et al., 2019*; *Andellini et al., 2015*; *Murphy and Fox, 2017*). However, GSR mathematically mandates that approximately 50% of correlations between regions will be negative (*Murphy and Fox, 2017*), thereby removing potentially meaningful differences in the proportion of anticorrelations; additionally, it has been shown across species and states of consciousness that the global signal contains information relevant for consciousness (*Tanabe et al., 2020*). Therefore, here we chose to avoid GSR in favour of the aCompCor denoising procedure, in line with previous work (*Luppi et al., 2019*; *Luppi et al., 2023c*; *Luppi et al., 2022a*).

Due to the presence of deformations caused by brain injury, rather than relying on automated pipelines, DOC patients' brains were individually preprocessed using SPM12, with visual inspections after each step. Additionally, to further reduce potential movement artefacts, data underwent despiking with a hyperbolic tangent squashing function, also implemented from the CONN toolbox (*Whitfield-Gabrieli and Nieto-Castanon, 2012*). The remaining preprocessing and denoising steps were the same as described above.

## Brain parcellation

Brains were parcellated into 454 cortical and subcortical regions of interest (ROIs). The 400 cortical ROIs were obtained from the scale-400 version of the recent Schaefer local-global functional parcellation (*Schaefer et al., 2018*). Since this parcellation only includes cortical regions, it was augmented with 54 subcortical ROIs from the highest resolution of the recent Tian parcellation (*Tian et al., 2020*). We refer to this 454-ROI parcellation as the 'augmented Schaefer' (*Luppi and Stamatakis, 2021*). To ensure the robustness of our results to the choice of atlas, we also replicated them using an alternative cortical parcellation of different dimensionality: we used the Schaefer scale-200 cortical parcellation, complemented with the scale-32 subcortical ROIs from the Tian subcortical atlas (*Luppi and Stamatakis, 2021*). The timecourses of denoised BOLD signals were averaged between all voxels belonging to a given atlas-derived ROI, using the CONN toolbox. The resulting region-specific timecourses of each subject were then extracted for further analysis in MATLAB.

## HRF deconvolution

In accordance with our previous work (*Luppi et al., 2022b*; *Luppi et al., 2023a*) and previous studies using of information-theoretic measures in the context of functional MRI data, we used a dedicated toolbox (*Wu et al., 2013*) to deconvolve the hemodynamic response function from our regional BOLD signal timeseries prior to analysis.

## Measuring integrated information

The framework of integrated information decomposition ($\Phi$ID) unifies integrated information theory (IIT) and partial information decomposition (PID) to decompose information flow into interpretable, disjoint parts. In this section we provide a brief description of $\Phi$ID and formulae required to compute the results. For further details, see *Mediano et al., 2021*; *Luppi et al., 2022b*.

## Partial information decomposition

We begin with Shannon's Mutual information (MI), which quantifies the interdependence between two random variables $X$ and $Y$. It is calculated as,

$$I(X;Y) = H(X)H(X|Y) = H(X) + H(Y)H(X,Y) \qquad (1)$$

where H($X$) stands for the Shannon entropy of a variable $X$. Above, the first equality states that the mutual information is equal to the reduction in entropy (i.e. uncertainty) about $X$ after $Y$ is known. Put simply, the mutual information quantifies the information that one variable provides about another (*Cover and Thomas, 2005*).

Crucially, *Williams and Beer, 2010* observed that the information that two source variables $X$ and $Y$ give about a third target variable $Z$, I($X,Y;Z$), should be decomposable in terms of different *types* of information: information provided by one source but not the other (unique information), by both sources separately (redundant information), or jointly by their combination (synergistic information). Following this intuition, they developed the *Partial Information Decomposition* (PID; *Williams and Beer, 2010*) framework, which leads to the following fundamental decomposition:

$$I(X,Y;Z) = Red(X,Y;Z) + Un(X;Z\backslash Y) + Un(Y;Z\backslash X) + Syn(X,Y;Z) \qquad (2)$$

Above, *Un* corresponds to the unique information that one source provides but the other doesn't, *Red* is the redundancy between both sources, and *Syn* is their synergy: information that neither $X$ nor $Y$ alone can provide, but that can be obtained by considering $X$ and $Y$ together.

The simplest example of a purely synergistic system is one in which $X$ and $Y$ are independent fair coins, and $Z$ is determined by the exclusive-OR function $Z$=XOR($X,Y$): that is, $Z$=0 whenever $X$ and $Y$ have the same value, and $Z$=1 otherwise. It can be shown that $X$ and $Y$ are both statistically independent of $Z$, which implies that neither of them provide – by themselves – information about $Z$. However, $X$ and $Y$ together fully determine $Z$, hence the relationship between $Z$ with $X$ and $Y$ is purely synergistic.

As another example for the case of Gaussian variables (as employed here), consider a 2-node coupled autoregressive process with two parameters: a noise correlation $c$ and a coupling parameter $a$. As $c$ increases, the system is flooded by 'common noise', making the system increasingly redundant because the common noise 'swamps' the signal of each node. As $a$ increases, each node has a stronger influence both on the other and on the system as a whole, and we expect synergy to increase. Therefore, synergy reflects the joint contribution of parts of the system to the whole that is not driven by common noise. This has been demonstrated through computational modelling (*Mediano et al., 2018*).

Recently, *Mediano et al., 2021* formulated an extension of PID able to decompose the information that multiple source variables have about multiple target variables. This makes PID applicable to the dynamical systems setting, and yields a decomposition with redundant, unique, and synergistic components in the past and future that can be used as a principled method to analyse information flow in neural activity (*Figure 3*).

## Synergy and redundancy calculation

While there is ongoing research on the advantages of different information decompositions for discrete data, most decompositions converge into the same simple form for the case of continuous Gaussian variables (*Barrett, 2015*). Known as *minimum mutual information PID* (MMI-PID), this decomposition quantifies redundancy in terms of the minimum mutual information of each individual source with the target; synergy, then, becomes identified with the additional information provided by the weaker source once the stronger source is known. Since linear-Gaussian models are sufficiently good descriptors of functional MRI timeseries and more complex, non-linear models offer no advantage (*Schulz et al., 2020*; *Nozari et al., 2024*), here we adopt the MMI-PID decomposition, following our own and others' previous applications of PID to neuroscientific data (*Luppi et al., 2022b*).

In a dynamical system such as the brain, one can calculate the amount of information flowing from the system's past to its future, known as time-delayed mutual information (TDMI). Specifically, by denoting the past of variables as $X_{t-\tau}$ and $Y_{t-\tau}$ and treating them as sources, and their joint future state ($X_t$, $Y_t$), as target, one can apply the PID framework and decompose the information flowing from past to future as

$$I\left(X_{t-\tau}, Y_{t-\tau}; X_t, Y_t\right) = \mathrm{Red}\left(X_{t-\tau}, Y_{t-\tau}; X_t, Y_t\right) + \mathrm{Un}\left(X_{t-\tau}; X_t, Y_t | Y_{t-\tau}\right) \\ + \mathrm{Un}\left(Y_{t-\tau}; X_t, Y_t \backslash X_{t-\tau}\right) + \mathrm{Syn}\left(X_{t-\tau}, Y_{t-\tau}; X_t, Y_t\right) \tag{3}$$

Applying $\Phi$ID to this quantity allows us to distinguish between redundant, unique, and synergistic information shared with respect to the future variables $X_t$, $Y_t$ (**Mediano et al., 2021**; **Luppi et al., 2022b**). Importantly, this framework, has identified $Syn\left(X_{t-\tau}, Y_{t-\tau}; X_t, Y_t\right)$ with the capacity of the system to exhibit emergent behaviour (**Mediano et al., 2022**) as well as a stronger notion of redundancy, in which information is shared by $X$ and $Y$ in both past and future. Accordingly, using the MMI-$\Phi$ID decomposition for Gaussian variables, we use

$$Red(X, Y) = \min\left\{I(X_{t-\tau}; X_t), I(X_{t-\tau}; Y_t), I(Y_{t-\tau}; X_t), I(Y_{t-\tau}; Y_t)\right\} \tag{4}$$

$$\mathrm{syn}(X, Y) = I(X_{t-\tau}, Y_{t-\tau}; X_t, Y_t) - \max\left\{I(X_{t-\tau}; X_t, Y_t), I(Y_{t-\tau}; X_t, Y_t)\right\} \tag{5}$$

Here, we used the Gaussian solver implemented in the JIDT toolbox (**Lizier, 2014**) to obtain TDMI, synergy and redundancy between each pair of brain regions, based on their HRF-deconvolved BOLD signal timeseries (**Mediano et al., 2021**; **Luppi et al., 2022b**).

## Revised measure of integrated information from Integrated Information Decomposition

Through the framework of Integrated Information Decomposition, we can decompose the constituent elements of $\Phi$, the formal measure of integrated information proposed by Integrated Information Theory to quantify consciousness (**Balduzzi and Tononi, 2008**). Note that several variants of $\Phi$ have been proposed over the years, including the original formulation of **Tononi, 2004**, other formulations based on causal perturbation (**Oizumi et al., 2014**; **Albantakis, 2022**) and others (see **Mediano et al., 2018**; **Tegmark, 2016** for comparative reviews). Here, we focus on the 'empirical $\Phi$' measure of Seth and Barrett (**Barrett and Seth, 2011**), based on the measures by **Balduzzi and Tononi, 2008** and adapted to applications to experimental data. It is computed as

$$\Phi = I\left(X_{t-\tau}, Y_{t-\tau}; X_t, Y_t\right) - I\left(X_{t-\tau}; X_t\right) - I\left(Y_{t-\tau}; Y_t\right) \tag{6}$$

and it quantifies how much temporal information is contained in the system over and above the information in its past. This measure is easy to compute (compared with other $\Phi$ measures) (**Oizumi et al., 2016**) and represents a noteworthy attempt to formalise the powerful intuitions underlying IIT. However, once the original formulation from Balduzzi and Tononi is rendered suitable for practical empirical application (**Barrett and Seth, 2011**; **Barrett and Mediano, 2019**) the resulting mathematical formulation has known shortcomings, including the fact that it can yield negative values in some cases – which are hard to interpret and seemingly paradoxical, as it does not seem plausible for a system to be 'negatively integrated' or an organism to have negative consciousness (**Barrett and Seth, 2011**; **Barrett and Mediano, 2019**).

Interestingly, with $\Phi$ID it can be formally demonstrated (**Mediano et al., 2021**) that $\Phi$ is composed of different information atoms: it contains all the synergistic information in the system, the unique information transferred from X to Y and vice versa, and, importantly, the subtraction of redundancy – which explains why $\Phi$ can be negative in redundancy-dominated systems.

To address this fundamental shortcoming, **Mediano et al., 2021** introduced a revised measure of integrated information, $\Phi_R$, which consists of the original $\Phi$ with the redundancy added back in:

$$\Phi_R = \Phi + Red(X, Y) \tag{7}$$

where Red(X, Y) is defined in *Equation (4)*. This measure is computationally tractable and preserves the original intuition of integrated information as measuring the extent to which 'the whole is greater than the sum of its parts', since it captures only synergistic and transferred information. Crucially, thanks to Integrated Information Decomposition, it can be proved that the improved formulation of integrated information that we adopt here is guaranteed to be non-negative (**Mediano et al., 2021**) – thereby avoiding a major conceptual limitation of the original formulation of $\Phi$.

Note that the formula for $\Phi^{WMS}$ above stems from what is known as IIT 2.0, but TDMI is by no means the only way of quantifying the dynamical structure of a system: indeed, subsequent developments in IIT 3.0 used alternative metrics with a more explicit focus on causal interpretations (**Oizumi et al.,**

*2014*), which were in turn replaced in the latest iteration known as IIT 4.0 (*Albantakis, 2022*; *Barbosa et al., 2021*). We do not consider the alternative measure of integrated information proposed in IIT 3.0 because it is computationally intractable for systems bigger than a small set of logic gates, and it is not universally well-defined (*Barrett and Mediano, 2019*).

## Gradient of redundancy-to-synergy relative importance to identify the synergistic workspace

After building networks of synergistic and redundant interactions between each pair of regions of interest (ROIs), we determined the role of each ROI in terms of its relative engagement in synergistic or redundant interactions. Following the procedure previously described (*Luppi et al., 2022b*), we first calculated the nodal strength of each brain region as the sum of all its interactions in the group-averaged matrix (*Figure 2—figure supplement 1*). Then, we ranked all 454 regions based on their nodal strength (with higher strength regions having higher ranks). This procedure was done separately for networks of synergy and redundancy. Subtracting each region's redundancy rank from its synergy rank yielded a gradient from negative (i.e. ranking higher in terms of redundancy than synergy) to positive (i.e. having a synergy rank higher than the corresponding redundancy rank; note that the sign is arbitrary).

It is important to note that the gradient is based on relative – rather than absolute – differences between regional synergy and redundancy; consequently, a positive rank difference does not necessarily mean that the region's synergy is greater than its redundancy; rather, it indicates that the balance between its synergy and redundancy relative to the rest of the brain is in favour of synergy – and vice versa for a negative gradient.

## Subdivision of workspace nodes into gateways and broadcasters

To identify which regions within the workspace play the role of gateways or broadcasters postulated in our proposed architecture, we followed a procedure analogous to the one adopted to identify the gradient of redundancy-synergy relative importance, but replacing the node *strength* with the node *participation coefficient*. The participation coefficient $P_i$ quantifies the degree of connection that a node entertains with nodes belonging to other modules: the more of a node's connections are towards other modules, the higher its participation coefficient will be (*Rubinov and Sporns, 2010*; *Rubinov and Sporns, 2011*). Conversely, the participation coefficient of a node will be zero if its connections are all with nodes belonging to its own module.

$$P_i = 1 - \sum_{s=1}^{M} \left( \frac{\kappa_{is}}{k_i} \right)^2 \tag{8}$$

Here, $\kappa_{is}$ is the strength of positive connections between node *i* and other nodes in module *s*, $k_i$ is the strength of all its positive connections, and *M* is the number of modules in the network. The participation coefficient ranges between zero (no connections with other modules) and one (equal connections to all other modules; *Rubinov and Sporns, 2010*; *Rubinov and Sporns, 2011*).

Here, modules were set to be the seven canonical resting-state networks identified by Yeo and colleagues (*Yeo et al., 2011*), into which the Schaefer parcellation is already divided (*Schaefer et al., 2018*), with the addition of an eighth subcortical network comprising all ROIs of the Tian subcortical network (*Tian et al., 2020*). The brain's RSNs were chosen as modules because of their distinct and well-established functional roles, which fit well with the notion of modules as segregated and specialised processing systems interfacing with the global workspace. Additionally, having the same definition of modules (i.e. RSNs) for synergy and redundancy allowed us to compute their respective participation coefficients in an unbiased way.

Separately for connectivity matrices of synergy and redundancy, the participation coefficient of each brain region was calculated. Then, regions belonging to the synergistic workspace were ranked, so that higher ranks indicated higher participation coefficient. Finally, the redundancy-based participation coefficient rank of each workspace region was subtracted from its corresponding synergy-based participation coefficient rank, to quantify – within the workspace – whether regions have relatively more diverse connectivity in terms of synergy, or in terms of redundancy.

This procedure yielded a gradient over workspace regions, from negative (i.e. having a more highly ranked participation coefficient based on redundancy than synergy) to positive (i.e. having a more highly ranked participation coefficient based on synergy than redundancy). Note that as before, the sign of this gradient is arbitrary, and it is based on relative rather than absolute difference. Workspace regions with a positive gradient value were classified as 'gateways', since they have synergistic interactions with many brain modules. In contrast, workspace regions with a negative value of the gradient – that is those whose redundancy rank is higher than their synergy rank, in terms of participation coefficient – were labelled as workspace 'broadcasters', since they possess information that is duplicated across multiple modules in the brain.

## Statistical Analysis

### Network-based statistic

The network-based statistic approach (*Zalesky et al., 2010*) was used to investigate the statistical significance of propofol-induced or DOC-induced alterations. This nonparametric statistical method is designed to control the family-wise error due to multiple comparisons, for application to graph data. Connected components of the graph are identified from edges that survive an a-priori statistical threshold (F-contrast; here we set the threshold to an F-value of 9, two-sided, with an alpha level of 0.05). In turn, the statistical significance of such connected components is estimated by comparing their topology against a null distribution of the size of connected components obtained from non-parametric permutation testing. This approach rejects the null hypothesis on a component-by-component level, and therefore achieves superior power compared to mass-univariate approaches (*Zalesky et al., 2010*).

### Testing for shared effects across datasets

We sought to detect changes that are common across datasets, to rule out possible propofol- or DOC-specific effects that are not related to consciousness per se (*Luppi et al., 2019*). To this end, we employed a null hypothesis significance test under the composite null hypothesis that *at least* one dataset among those considered here has no effect. In other words, for the null hypothesis to be rejected we demand that all comparisons exhibit non-zero effects. As usual, the test proceeds by comparing an observed test statistic with a null distribution. The test statistic is the minimum of the three F-scores obtained in the comparisons of interest (DOC vs awake; anaesthesia vs awake; and anaesthesia vs recovery), and the null distribution is sampled by randomly reshuffling exactly one dataset (picked at random) at a time and recalculating the F-scores. By shuffling exactly one dataset (instead of all of them), we are comparing the observed data against the 'least altered' version of the data that is still compatible with the null hypothesis. This is a type of least favourable configuration test (*Lehmann and Romano, 2005*), which is guaranteed to control the false positive rate below a set threshold (here, 0.05). The details of this test will be described in a future publication. Common changes across the three states of consciousness were then identified as edges (defined in terms of $\Phi_R$) that were either (i) increased in DOC compared with control; (ii) increased during anaesthesia compared with wakefulness; and (iii) increased during anaesthesia compared with post-anaesthetic recovery; or (i) decreased in DOC compared with control; (ii) decreased during anaesthesia compared with wakefulness; and (iii) decreased during anaesthesia compared with post-anaesthetic recovery.

### Spatial autocorrelation-preserving null model for correlation

The significance of correlation between nodes' participation coefficient based on different definitions of modules (a-priori as resting-state networks or in a data-driven fashion from Louvain community detection) was assessed using a spatial permutation test which generates a null distribution of 10,000 randomly rotated brain maps with preserved spatial covariance ('spin test'), to ensure robustness to the potential confounding effects of spatial autocorrelation (*Markello and Misic, 2021*; *Váša and Mišić, 2022*; *Alexander-Bloch et al., 2018*).

## Code availability

The Java Information Dynamics Toolbox v1.5 is freely available online: (https://github.com/jlizier/jidt; *Lizier et al., 2018*). The CONN toolbox version 17 f is freely available online (http://www.nitrc.org/projects/conn). The Brain Connectivity Toolbox code used for graph-theoretical analyses is freely

available online (https://sites.google.com/site/bctnet/). The HRF deconvolution toolbox v2.2 is freely available online: (https://www.nitrc.org/projects/rshrf). The code for spin-based permutation testing of cortical correlations is freely available at https://github.com/frantisekvasa/rotate_parcellation (*Váša, 2023*). We have made freely available MATLAB/Octave and Python code to compute measures of Integrated Information Decomposition of timeseries with the Gaussian MMI solver, at https://github.com/Imperial-MIND-lab/integrated-info-decomp (*Liu and Luppi, 2023*).

## Acknowledgements

Author AIL is grateful to Dr. Athena Demertzi and Dr. Petra Vertes for helpful discussion. This work was supported by the Gates Cambridge Trust (OPP1144; to AIL); the Wellcome Trust (grant no. 210920/Z/18/Z, PAM and DB); grants from the UK Medical Research Council [U.1055.01.002.00001.01 to AMO and JDP]; the Canadian Institute for Advanced Research (CIFAR; grant RCZB/072 RG93193) [to DKM and EAS]; the James S McDonnell Foundation [to AMO and JDP]; and the Canada Excellence Research Chairs program (215063 to AMO); the National Institute for Health Research (NIHR, UK), Cambridge Biomedical Research Centre and NIHR Senior Investigator Awards [to DKM], the Stephen Erskine Fellowship (Queens' College, Cambridge, to EAS), the L'Oreal-Unesco for Women in Science Excellence Research Fellowship to LN; the British Oxygen Professorship of the Royal College of Anaesthetists [to DKM]; FR is funded by the Ad Astra Chandaria Foundation. The research was also supported by the NIHR Brain Injury Healthcare Technology Co-operative based at Cambridge University Hospitals NHS Foundation Trust and University of Cambridge. AMO and DKM are Fellows of the CIFAR Brain, Mind, and Consciousness Programme. Data were provided [in part] by the Human Connectome Project, WU-Minn Consortium (Principal Investigators: David Van Essen and Kamil Ugurbil; 1U54MH091657) funded by the 16 NIH Institutes and Centers that support the NIH Blueprint for Neuroscience Research; and by the McDonnell Center for Systems Neuroscience at Washington University.

## Additional information

### Competing interests

Michael M Craig: currently employed by Valence Labs. The work contributing to the manuscript was performed as part of his graduate studies at the University of Cambridge, and is in no way related to his employment at Valence Labs. The other authors declare that no competing interests exist.

### Funding

| Funder | Grant reference number | Author |
|---|---|---|
| Gates Cambridge Trust | OPP1144 | Andrea I Luppi |
| Wellcome Trust | 10.35802/210920 | Daniel Bor<br>Pedro AM Mediano |
| Medical Research Council | U.1055.01.002.00001.01 | John Pickard<br>Adrian M Owen |
| James S. McDonnell Foundation | | John Pickard<br>Adrian M Owen |
| Canada Excellence Research Chairs, Government of Canada | 215063 | Adrian M Owen |
| Queens' College Cambridge | Stephen Erskine Fellowship | Emmanuel A Stamatakis |
| Fondation L'Oréal | Women in Science Excellence Research Fellowship | Lorina Naci |

| Funder | Grant reference number | Author |
|---|---|---|
| UNESCO | Women in Science Excellence Research Fellowship | Lorina Naci |
| Royal College of Anaesthetists | British Oxygen Professorship | David K Menon |
| Ad Astra Chandaria Foundation | | Fernando E Rosas |
| Canadian Institute for Advanced Research | RCZB/072 RG93193 | David K Menon Emmanuel A Stamatakis |

The funders had no role in study design, data collection and interpretation, or the decision to submit the work for publication. For the purpose of Open Access, the authors have applied a CC BY public copyright license to any Author Accepted Manuscript version arising from this submission.

## Author contributions

Andrea I Luppi, Conceptualization, Formal analysis, Investigation, Visualization, Methodology, Writing – original draft; Pedro AM Mediano, Conceptualization, Software, Methodology, Writing – review and editing; Fernando E Rosas, Conceptualization, Methodology, Writing – review and editing; Judith Allanson, Data curation, Investigation, Project administration, Writing – review and editing; John Pickard, Funding acquisition, Project administration, Writing – review and editing; Robin L Carhart-Harris, Supervision, Writing – review and editing; Guy B Williams, Investigation, Writing – original draft; Michael M Craig, Lorina Naci, Data curation, Investigation, Writing – review and editing; Paola Finoia, Investigation, Writing – review and editing; Adrian M Owen, Funding acquisition, Investigation, Writing – review and editing; David K Menon, Supervision, Funding acquisition, Project administration; Daniel Bor, Conceptualization, Supervision, Writing – review and editing; Emmanuel A Stamatakis, Supervision, Funding acquisition, Project administration, Writing – review and editing

## Author ORCIDs

Andrea I Luppi https://orcid.org/0000-0002-3461-6431
Fernando E Rosas https://orcid.org/0000-0001-7790-6183
Lorina Naci https://orcid.org/0000-0001-9630-3978
Daniel Bor https://orcid.org/0000-0002-0741-8157
Emmanuel A Stamatakis https://orcid.org/0000-0001-6955-9601

## Ethics

Anaesthesia data: The study received ethical approval from the Health Sciences Research Ethics Board and Psychology Research Ethics Board of Western University (Ontario, Canada). DOC data: Ethical approval for this study was provided by the National Research Ethics Service (National Health Service, UK; LREC reference 99/391). Each volunteer provided written informed consent, and received monetary compensation for their time.

Reviewer #2 (Public Review): https://doi.org/10.7554/eLife.88173.4.sa1
Reviewer #3 (Public Review): https://doi.org/10.7554/eLife.88173.4.sa2
Author response https://doi.org/10.7554/eLife.88173.4.sa3

# Additional files

## Supplementary files
• MDAR checklist

## Data availability

The Human Connectome Project (HCP) dataset is available on ConnectomeDB (https://www.humanconnectome.org/study/hcp-young-adult/data-releases). Due to patient privacy concerns, DOC patient data are available upon request by qualified researchers, for non-commercial use only. The UK Health Research Authority mandates that the confidentiality of data is the responsibility

of Chief Investigators for the initial studies (in this case, Dr. Allanson and Prof Menon; and anyone to whom this responsibility is handed - for example, in the context of retirement or transfer to another institution). For researchers interested in working with this dataset, please contact the Data Access Committee: Dr. Judith Allanson (judith.allanson.1@gmail.com), Prof. David Menon ( dkm13@cam.ac.uk) or Dr. Emmanuel Stamatakis (eas46@cam.ac.uk). The request should include an exact description of the specific planned analysis, the name of the academic institution and department, and the name of the researcher who would bear responsibility. Requests will be considered to assess the feasibility and appropriateness of the proposed study, and the capacity to maintain the required levels of data security, consistent with the original approved Research Ethics approval, and the patient information sheet that was the basis of consent obtained. In case of approval, recipients of the data will be required to a legal agreement confirming that the data will only be used as laid out in the approved proposal, confirming that the data will be protected and not be shared further, and confirming that the data will be deleted as soon as the project finishes. The propofol dataset is available on the OpenNeuro data repository (https://doi.org/10.18112/open-neuro.ds003171.v2.0.1).

The following previously published datasets were used:

| Author(s) | Year | Dataset title | Dataset URL | Database and Identifier |
|---|---|---|---|---|
| Kandeepan S, Jorge R, Francisco G, Bobby S, Sreeram V, Mark OA, Lorina N, Andrea S, Sophia NE | 2020 | Modeling an auditory stimulated brain under altered states of consciousness using the generalized ising model | https://doi.org/10.18112/openneuro.ds003171.v2.0.1 | OpenNeuro, 10.18112/openneuro.ds003171.v2.0.1 |
| Van Essen DC, Ugurbil K, Auerbach E, Barch D, Behrens TEJ, Bucholz R, Chang A, Chen L, Corbetta M, Curtiss SW, Della Penna S, Feinberg D, Glasser MF, Harel N, Heath AC, Larson-Prior L, Marcus D, Michalareas G, Moeller S, Oostenveld R, Petersen SE, Prior F, Schlaggar BL, Smith SM, Snyder AZ, Xu J, Yacoub E | 2017 | Human Connectome Young Adult - 1200 Subjects Data Release | https://www.humanconnectome.org/study/hcp-young-adult/data-releases | Human Connectome Project, 1200 |

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
