## [Editor Report · eLife assessment]

This article presents **important** results describing how the gathering, integration, and broadcasting of information in the brain changes when consciousness is lost either through anesthesia or injury. They provide **convincing** evidence to support their conclusions, although the paper relies on a single analysis tool (partial information decomposition) and could benefit from a clearer explication of its conceptual basis, methodology, and results. The work will be of interest to both neuroscientists and clinicians interested in basic and clinical aspects of consciousness.

---

## [Referee Report · Reviewer #2 (Public Review)]

The authors analysed functional MRI recordings of brain activity at rest, using state-of-the-art methods that reveal the diverse ways in which information can be integrated in the brain. In this way, they found brain areas that act as (synergistic) gateways for the 'global workspace', where conscious access to information or cognition would occur, and brain areas that serve as (redundant) broadcasters from the global workspace to the rest of the brain. The results are compelling and are consistent with the already assumed role of several networks and areas within the Global Neuronal Workspace framework. Thus, in a way, this work comes to stress the role of synergy and redundancy as complementary information processing modes, which fulfill different roles in the bigger context of information integration.

In addition, to prove that the identified high-order interactions are relevant to the phenomenon of consciousness, the same analysis was performed in subjects under anesthesia or with disorders of consciousness (DOC), showing that indeed the loss of consciousness is associated with a deficient integration of information within the gateway regions.

---

## [Referee Report · Reviewer #3 (Public Review)]

The work proposes a model of neural information processing based on a 'synergistic global workspace,' which processes information in three principal steps: a gatekeeping step (information gathering), an information integration step, and finally, a broadcasting step. They provided an interpretation of the reduced human consciousness states in terms of the proposed model of brain information processing, which could be helpful to be implemented in other states of consciousness. The manuscript is well-organized, and the results are important and could be interesting for a broad range of literature, suggesting interesting new ideas for the field to explore.

---

## [Author Response]

The following is the authors’ response to the previous reviews.

We are pleased that Reviewer 3 has deemed our revisions satisfactory; below, we provide responses to the remaining Recommendations for the Authors from Reviewer 2.

**Reviewer #2 (Recommendations For The Authors):**
Minor corrections:Line 91: GWT should be GNWT

Fixed, thank you.

Figure 2: fix the label "Participationcoefficient rank" (no space between Participation and coefficient)

Fixed, thank you for spotting.

Line 317: Figure 2 should be Figure 3

Fixed, thank you.

Line 360: Figure 4D, right?

Fixed, thank you. We also confirm that Figure 4 and its caption are correct. Under anaesthesia, many regions have more Integrated Information than during Recovery (red regions), but the only changes that are consistently observed across all three contrasts are the decreases.

Line 375: Should be Figure 5A

Fixed, thank you.

The recovery period of the anesthesia data is not described in Methods.

We have now added the missing information:

“Propofol was discontinued following the deep anaesthesia scan, and participants reached level 2 of the Ramsey scale approximately 11 minutes afterwards, as indicated by clear and rapid responses to verbal commands. This corresponds to the “recovery” period 176.”

We have also expanded our discussion on the interaction between information decomposition and measures of directionality:

“Indeed, transfer entropy can itself be decomposed into information-dynamic atoms through Partial Information Decomposition and Integrated Information Decomposition 33,34,49,151; ΦID can further decompose the Normalised Directed Transfer Entropy measure used by Deco et al 5, as recently demonstrated 152. We look forward to a more refined conceptualization of the synergistic workspace architecture that takes into account both information types and the directionality of information flow – especially in datasets with higher temporal resolution.”